# Perturbation-Induced Linearization: Constructing Unlearnable Data with Solely Linear Classifiers

**Jinlin Liu, Wei Chen, Xiaojin Zhang**[*]
Huazhong University of Science and Technology, Wuhan 430074, China
`{jinlinliu, lemuria_chen, xiaojinzhang}@hust.edu.cn`

## Abstract

Collecting web data to train deep models has become increasingly common, raising concerns about unauthorized data usage. To mitigate this issue, unlearnable examples introduce imperceptible perturbations into data, preventing models from learning effectively. However, existing methods typically rely on deep neural networks as surrogate models for perturbation generation, resulting in significant computational costs. In this work, we propose Perturbation-Induced Linearization (PIL), a computationally efficient yet effective method that generates perturbations using only linear surrogate models. PIL achieves comparable or better performance than existing surrogate-based methods while reducing computational time dramatically. We further reveal a key mechanism underlying unlearnable examples: inducing linearization to deep models, which explains why PIL can achieve competitive results in a very short time. Beyond this, we provide an analysis about the property of unlearnable examples under percentage-based partial perturbation. Our work not only provides a practical approach for data protection but also offers insights into what makes unlearnable examples effective. Code is available at https://github.com/jinlinll/pil.

## 1 Introduction

Collecting vast amounts of data from the Internet has become a common practice for training advanced deep learning models (Russakovsky et al., 2015; Zhang et al., 2018). However, much of this data—including human faces (Birhane & Prabhu, 2021), artwork, and text—is scraped without the consent of the original creators, raising serious concerns about the unauthorized use of personal data. To mitigate this issue, researchers have proposed *unlearnable examples*, a family of data protection methods that add imperceptible perturbations to the data. The perturbed data look unchanged to humans, but when used for training, they cause deep neural networks (DNNs) to fail to generalize and perform like random guessing on unseen samples. The intuition is that by rendering scraped data "unlearnable," unauthorized third parties will be disincentivized from exploiting it for model training.

Existing studies often use deep neural networks (DNNs) as surrogates to generate such protective perturbations. However, these approaches are often computationally intensive, as training complex surrogate DNNs and executing adversarial attack methods like PGD (Madry et al., 2017) are very time consuming. For instance, the REM (Fu et al., 2022) method needs over 15 GPU hours to generate perturbations for the CIFAR-10 dataset.

In this paper, we introduce **Perturbation-Induced Linearization (PIL)**, a novel method for degrading the generalization ability of DNNs. PIL creates perturbation–label correspondences that can be easily captured by simple linear models, thereby inducing the linearization of DNNs. PIL employs a simple **linear surrogate model** to generate perturbations that transfer effectively to various deep learning models. Owing to the simplicity of linear models, PIL is highly efficient, for example, it requires less than one GPU minute to generate perturbations for the CIFAR-10 dataset. Moreover, we demonstrate PIL do improve the linearity of deep models and find that existing unlearnable example methods, though not designed to induce linearization, also cause deep models to exhibit stronger linear behavior. This suggests that induced linearization may be the underlying mechanism behind the success of unlearnable examples.

---

[*]Corresponding author

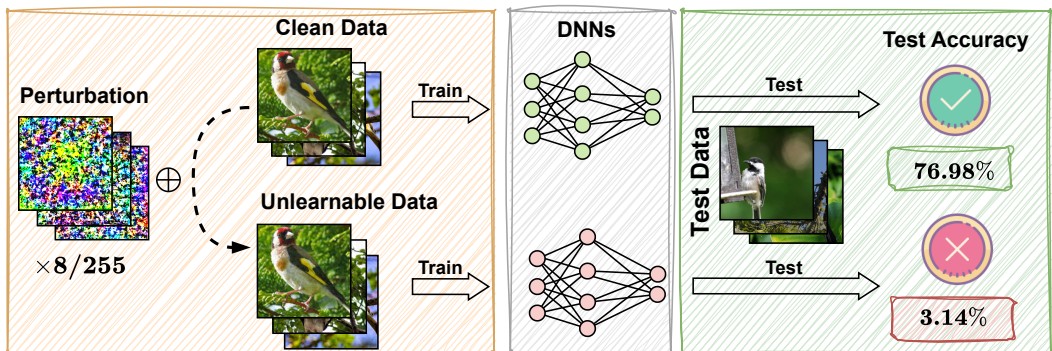

Figure 1: Illustration of the workflow of unlearnable examples. DNNs trained on the unlearnable data perform poorly on the clean test data. Results are reported for PIL on ImageNet-100.

We further conduct comprehensive experiments to evaluate the effectiveness of PIL and to analyze the performance and fundamental property of unlearnable examples under percentage-based partial perturbation. Our contributions can be summarized as follows:

- We propose PIL, an efficient method for generating protective perturbations using a simple linear surrogate model, and demonstrate its effectiveness across architectures, datasets, and defenses.

- We uncover a key mechanism behind unlearnable examples: they induce deep models to behave more like linear models, which may reduce their capacity to learn meaningful representations.

- We provide an analysis revealing a fundamental property of unlearnable examples: they cannot substantially reduce test accuracy when only part of the dataset is perturbed.

## 2 RELATED WORK

**Unlearnable Examples.** Unlearnable examples (Huang et al., 2021; Jiang et al., 2023; Zhu et al., 2024; Hapuarachchi et al., 2024; Wang et al., 2024), also referred to as availability attacks (Fowl et al., 2021; Yu et al., 2022), perturbative availability poisons (Liu et al., 2023), generalization attacks (Yuan & Wu, 2021), or delusive attacks (Tao et al., 2021), aim to protect datasets from unauthorized exploitation. One of the earliest unlearnable examples (Biggio et al., 2012) considered convex models and demonstrated that optimizing a single poisoning sample could significantly disrupt SVM training. However, extending poisoning techniques to DNNs is much more challenging. Several methods have recently been proposed. Error-Minimizing (EM) (Huang et al., 2021) attacks minimize the classification errors of images while training a surrogate deep learning model. Robust Error-Minimizing (REM) (Fu et al., 2022) replaces the normally-trained surrogate in EM with an adversarially-trained model, making the attacks more robust against adversarial training. Targeted Adversarial Poisoning (TAP) (Fowl et al., 2021) further extends this idea by using a fixed, pretrained surrogate model and minimizing the classification loss with target labels. Neural Tangent Generalization Attacks (NTGA) (Yuan & Wu, 2021) use the neural tangent kernel (Jacot et al., 2018) as the surrogate to model the training dynamics of a class of wide DNNs. In addition, several works have explored using multiple models as surrogates (Chen et al., 2022) or developing surrogate-free approaches (Yu et al., 2022; Sandoval-Segura et al., 2022; Sadasivan et al., 2023). It is worth noting that the evaluation criteria of unlearnability is not yet unified. We provide a detailed discussion in Appendix A.1.

**Adversarial Attacks.** Adversarial attacks (Szegedy et al., 2013; Goodfellow et al., 2014; Kurakin et al., 2018; Carlini & Wagner, 2017; Kurakin et al., 2018; Carlini & Wagner, 2017; Madry et al., 2017; Croce & Hein, 2020) typically craft perturbations that maximize the model's prediction loss. Existing research has demonstrated that adversarial attacks can effectively deceive deep neural networks at the inference stage. However, conventional adversarial examples, such as those generated by a 20-step Projected Gradient Descent (PGD-20) attack, fail to mislead deep neural networks during the training phase like unlearnable examples (Huang et al., 2021). Further investigations (Fowl et al., 2021) reveal that increasing the steps of PGD attack to 250 (PGD-250) enables the generated adversarial examples to also deceive deep neural networks during training.

**Shortcut learning.** Shortcut learning (Geirhos et al., 2020) refers to the phenomenon where deep neural networks rely on spurious correlations or low-level cues in the training data rather than learning the intended, generalizable representations. This phenomenon is common in deep learning, appearing in tasks like image classification (Beery et al., 2018), where models rely on background features, and question answering (Niven & Kao, 2019), where they exploit superficial textual cues. Recent work (Yu et al., 2022) suggests that unlearnable examples essentially embed imperceptible shortcuts. In particular, perturbations generated by many existing methods can be recognized by simple linear models. Motivated by this observation, we design perturbations that linear models can easily associate with class labels.

## 3 THE PROPOSED METHOD

### 3.1 PROBLEM STATEMENT

We frame the generation of unlearnable examples within a standard attacker-defender framework, which we outline below.

**Threat Model.** The *defender* is the data owner (e.g., a user posting personal photos, an artist sharing their work) who wishes to prevent their data from being used for unauthorized model training. Before releasing the data, the defender adds carefully crafted, imperceptible perturbations. The *attacker* is an unauthorized party who scrapes this publicly available, perturbed data to train a deep learning model. The attacker is assumed to have full access to the perturbed dataset but not the original clean data or the perturbations. The defender's ultimate goal is to make any model trained by the attacker generalize poorly to clean, unseen data, thereby disincentivizing the unauthorized use of their data. Our proposed method, PIL, is a defense mechanism from this perspective.

**Notation and Objective.** We consider standard image classification with a DNN $f_\theta$. Let the clean training set be $\mathcal{D}_c = \{(\boldsymbol{x}_i, y_i)\}_{i=1}^n$ with $\boldsymbol{x}_i \in \mathbb{R}^d$ and labels $y_i \in \{1, \ldots, K\}$. The defender constructs an unlearnable dataset $\mathcal{D}_u = \{(\boldsymbol{x}_i', y_i)\}_{i=1}^n$ by inducing imperceptible perturbations $\boldsymbol{x}_i' = \boldsymbol{x}_i + \boldsymbol{\delta}_i$ subject to $\|\boldsymbol{\delta}_i\|_p \leq \epsilon$. In this paper, we specifically use $\|\boldsymbol{\delta}_i\|_\infty \leq 8/255$.

The defender's objective is to craft perturbations $\{\boldsymbol{\delta}_i\}_{i=1}^n$ that degrade the generalization performance of any model $f_{\theta^*}$ trained on the unlearnable dataset $\mathcal{D}_u$. This goal can be formalized as a bilevel optimization problem where the defender aims to maximize the final test loss of the attacker's model:

$$\max_{\{\boldsymbol{\delta}_i\}_{i=1}^n} \quad \mathbb{E}_{(\boldsymbol{x}_i, y_i) \sim \mathcal{D}_t}[\ell(f_{\theta^*}(\boldsymbol{x}_i), y_i)]$$
$$\text{s.t.} \quad \theta^* = \arg\min_\theta \mathbb{E}_{(\boldsymbol{x}_i, y_i) \sim \mathcal{D}_c}[\ell(f_\theta(\boldsymbol{x}_i + \boldsymbol{\delta}_i), y_i)]. \tag{1}$$

Here, $\ell(\cdot, \cdot)$ is the loss function (typically cross-entropy), and $\mathcal{D}_t$ is the clean test set. The inner optimization problem describes the attacker's training process on the perturbed data, while the outer optimization represents the defender's goal of maximizing test error.

### 3.2 PERTURBATION-INDUCED LINEARIZATION

Our core idea is to design a perturbation $\boldsymbol{\delta}$ that forces deep neural networks to ignore the complex semantic features of the original image $\boldsymbol{x}$ during training, and instead learn a simple **linear mapping** between $\boldsymbol{\delta}$ and the label $y$. To achieve this, we employ a bias-free linear classifier $f_{\text{lin}}$ to guide the perturbation generation:

$$f_{\text{lin}}(\boldsymbol{x}; \boldsymbol{w}) = \boldsymbol{x}\boldsymbol{w}, \quad \boldsymbol{w} \in \mathbb{R}^{d \times k}, \tag{2}$$

where $k$ denotes the number of classes.

We aim to optimize a perturbation set $\{\boldsymbol{\delta}_i = \boldsymbol{\delta}_i^1 + \boldsymbol{\delta}_i^2\}_{i=1}^n$, where each perturbation $\boldsymbol{\delta}_i$ for a training sample $(\boldsymbol{x}_i, y_i)$ conceptually consists of two components with distinct objectives:

1. **Semantic Obfuscation.** We require that the main semantic content of $\boldsymbol{x}_i$ carries little useful information. This is enforced by encouraging the prediction on the obfuscated image $\boldsymbol{x}_i - \boldsymbol{\delta}_i^1$ to be close to a uniform distribution, which is implemented by minimizing the KL divergence:

$$\{\boldsymbol{\delta}_i^1\}_{i=1}^n = \arg\min_{\{\boldsymbol{\delta}_i\}_{i=1}^n} \mathbb{E}_{(\boldsymbol{x}_i, y_i) \sim \mathcal{D}_c}\left[L_{\text{KL}}\left(f_{\text{lin}}(\boldsymbol{x}_i - \boldsymbol{\delta}_i; \boldsymbol{w}), \tfrac{1}{k}\mathbf{1}_{1 \times k}\right)\right]. \tag{3}$$

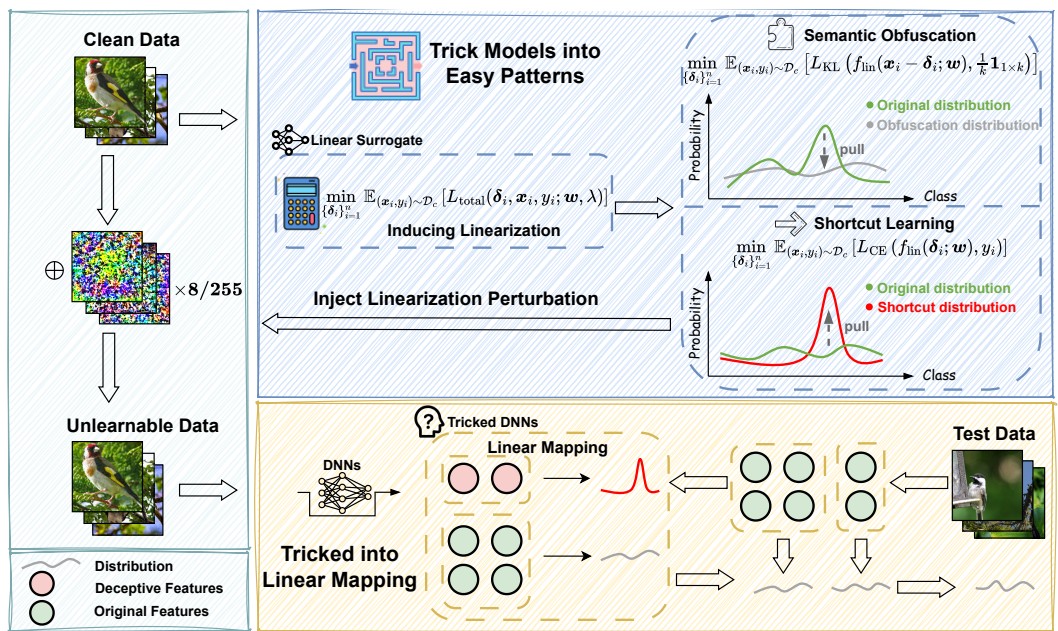

Figure 2: Architecture illustration of PIL. We use $\oplus$ to denote inducing perturbations into images. Best viewed in color. Zoom in for details.

2. **Shortcut Learning.** We further require $\boldsymbol{\delta}_i^2$ itself to encode strong class-specific signals, serving as an easily learnable shortcut. This is achieved by enforcing that $f_{\mathrm{lin}}$ can accurately predict $y_i$ directly from $\boldsymbol{\delta}_i^2$, via cross-entropy minimization:

$$\{\boldsymbol{\delta}_i^2\}_{i=1}^n = \arg \min_{\{\boldsymbol{\delta}_i\}_{i=1}^n} \mathbb{E}_{(\boldsymbol{x}_i, y_i) \sim \mathcal{D}_c} \left[ L_{\mathrm{CE}} \left( f_{\mathrm{lin}}(\boldsymbol{\delta}_i; \boldsymbol{w}), y_i \right) \right]. \tag{4}$$

These two components, $\boldsymbol{\delta}_i^1$ and $\boldsymbol{\delta}_i^2$, are only a conceptual decomposition used to define the objectives. In practice, we optimize a single perturbation $\boldsymbol{\delta}_i$ to jointly satisfy both goals. To this end, we combine the objectives into a unified loss function:

$$L_{\mathrm{total}}(\boldsymbol{\delta}, \boldsymbol{x}, y; \boldsymbol{w}, \lambda) = \lambda L_{\mathrm{CE}} \left( f_{\mathrm{lin}}(\boldsymbol{\delta}; \boldsymbol{w}), y \right) + (1 - \lambda) L_{\mathrm{KL}} \left( f_{\mathrm{lin}}(\boldsymbol{x} - \boldsymbol{\delta}; \boldsymbol{w}), \frac{1}{k} \mathbf{1}_{1 \times k} \right). \tag{5}$$

where $\lambda$ is a balance parameter controlling the trade-off between shortcut learning and semantic obfuscation.

The optimal perturbations are then obtained by minimizing the total loss across the dataset with norm constraints (e.g., $\|\boldsymbol{\delta}_i\|_\infty \leq \epsilon$) to ensure imperceptibility:

$$\{\boldsymbol{\delta}_i^*\}_{i=1}^n = \arg \min_{\{\boldsymbol{\delta}_i\}_{i=1}^n} \mathbb{E}_{(\boldsymbol{x}_i, y_i) \sim \mathcal{D}_c} \left[ L_{\mathrm{total}}(\boldsymbol{\delta}_i, \boldsymbol{x}_i, y_i; \boldsymbol{w}, \lambda) \right]. \tag{6}$$

Finally, the unlearnable dataset is constructed as:

$$\mathcal{D}_u = \{(\boldsymbol{x}_i - \boldsymbol{\delta}_i^*, y_i)\}_{i=1}^n. \tag{7}$$

It is worth noting that we subtract $\boldsymbol{\delta}_i^*$ from the original image, which ensures that when the target model $f_\theta$ behaves approximately linearly, its output can be decomposed as:

$$f_{\mathrm{lin}}(\boldsymbol{x}_i - \boldsymbol{\delta}_i^*; \boldsymbol{w}) = f_{\mathrm{lin}}(\boldsymbol{x}_i - \boldsymbol{\delta}_i^{1^*}; \boldsymbol{w}) + f_{\mathrm{lin}}(-\boldsymbol{\delta}_i^{2^*}; \boldsymbol{w}). \tag{8}$$

According to our optimization goals, the first term approaches a uniform distribution (low information), while the second term is strongly correlated with the label $y_i$. This design enforces $f_\theta$ to capture the negative correlation between $\boldsymbol{\delta}^*$ and $y$, rather than the semantic relation between $\boldsymbol{x}$ and $y$.

Algorithm 1 outlines the main steps of PIL. In Lines 1–4 the bias-free linear model $f_{\mathrm{lin}}$ is trained on the dataset $\mathcal{D}$ using SGD. In Lines 5–8 the trained linear surrogate is employed to optimize the perturbations: specifically, Lines 6–8 describe a PGD-like procedure for updating each perturbation

---

**Algorithm 1:** Perturbation-Induced Linearization

---

**Input:** Initial perturbation $\{\boldsymbol{\delta}_i\}_{i=1}^n$, model $f_{\text{lin}}(\cdot; \boldsymbol{w})$, dataset $\mathcal{D}$, learning rate $\eta$, perturbation
       update rate $\alpha$, perturbation budget $\epsilon$, balancing factor $\lambda$, number of iterations $M$ and $N$
**Output:** Optimized $\{\boldsymbol{\delta}_i\}_{i=1}^n$

1 **for** $m$ *in* $1 \cdots M$ **do**
2     **for** $\boldsymbol{x}, y$ *in* $\mathcal{D}$ **do**
3        Compute gradient $\nabla_{\boldsymbol{w}} L_{\text{CE}}(f_{\text{lin}}(\boldsymbol{x}; \boldsymbol{w}), y)$;
4        Update $\boldsymbol{w} \leftarrow \boldsymbol{w} - \eta \nabla_{\boldsymbol{w}} L_{\text{CE}}(f_{\text{lin}}(\boldsymbol{x}; \boldsymbol{w}), y)$;

5 **for** $\boldsymbol{x}_i, y_i$ *in* $\mathcal{D}$ **do**
6     **for** $n$ *in* $1 \cdots N$ **do**
7        Compute gradient $\nabla_{\boldsymbol{\delta}_i} L_{\text{total}}(\boldsymbol{\delta}_i, \boldsymbol{x}_i, y_i; \boldsymbol{w}, \lambda)$;
8        Update $\boldsymbol{\delta}_i \leftarrow \text{Clip}(\boldsymbol{\delta}_i - \alpha \cdot \text{sign}(\nabla_{\boldsymbol{\delta}_i} L_{\text{total}}(\boldsymbol{\delta}_i, \boldsymbol{x}_i, y_i; \boldsymbol{w}, \lambda)), -\epsilon, \epsilon)$;

9 **return** $\{\boldsymbol{\delta}_i\}_{i=1}^n$

---

$\boldsymbol{\delta}_i$. To ensure the perturbation remains visually imperceptible, it is clipped after each update by $\|\boldsymbol{\delta}_i\|_\infty \leq \epsilon$. Once the optimized perturbations are obtained, the unlearnable dataset is constructed as in Eq. 7. Importantly, we believe that pretraining the linear surrogate on the dataset enhances the semantic obfuscation defined in Eq. 3. Without pretraining, a randomly initialized model has no knowledge of the training data, making it difficult to optimize for semantic obfuscation. In contrast, with a pretrained surrogate, the semantic structure of the data is better captured, and our experiments confirm that PIL achieves stronger protection under this setting (see Appendix A.7).

## 4 EXPERIMENTS

**Datasets and Models.** We evaluate our method on four widely used image classification benchmarks: SVHN (Netzer et al., 2011), CIFAR-10, CIFAR-100 (Krizhevsky et al., 2009), and a 100-class subset[1] of ImageNet (Russakovsky et al., 2015). In particular, the experiments on the ImageNet subset are designed to verify the effectiveness of our method on high-resolution images. To validate the architecture-independent property of our method, we conduct experiments across diverse neural network architectures, including ResNet-18, ResNet-50 (He et al., 2016), VGG-19 (Simonyan & Zisserman, 2014), DenseNet-121 (Huang et al., 2017), and MobileNet-V2 (Sandler et al., 2018). By default, we use the CIFAR-10 dataset and ResNet-18 model unless otherwise specified.

**Experimental Setting for PIL.** We initialize perturbations $\{\boldsymbol{\delta}_i\}_{i=1}^n$ from a uniform distribution Uniform$(-\epsilon, \epsilon)$ with budget $\epsilon = 8/255$. Updates are performed with step size $\alpha = 8/2550$, and the balancing factor[2] is set to $\lambda = 0.9$. Further hyperparameter details are provided in Appendix A.5.

**Experimental Setting for Baselines.** We compare our method with several representative unlearnable example baselines, including surrogate-based methods: Error-Minimizing (EM) (Huang et al., 2021), Robust Error-Minimizing (REM) (Fu et al., 2022), Targeted Adversarial Poisoning (TAP) (Fowl et al., 2021), Neural Tangent Generalization Attacks (NTGA) (Yuan & Wu, 2021), Self-Ensemble Protection (SEP) (Chen et al., 2022), and surrogate-free methods: Synthetic Perturbations (SP) (Yu et al., 2022), AutoRegressive poisoning (AR) (Sandoval-Segura et al., 2022), and Convolution-based Unlearnable Datasets (CUDA) (Sadasivan et al., 2023). Detailed descriptions of these baseline methods can be found in Appendix A.6. For visualization of unlearnable examples generated by different methods, please refer to Appendix A.13.

**Experimental Setting for Testing.** All experiments follow the same procedure: we train the network on the unlearnable dataset for 100 epochs and then evaluate its accuracy on a clean test set. Training use an initial learning rate of 0.1, cosine annealing learning rate schedule, weight decay of $1 \times 10^{-4}$, and momentum of 0.9. For experiments on the ImageNet Subset, we additionally apply gradient clipping with a threshold of 1.0. Details of computer resources can be found in Appendix A.14.

**Perturbation Norm.** According to previous studies, an $L_\infty$ perturbation with $\epsilon = 8/255$ is commonly considered imperceptible (Huang et al., 2021). Our method (PIL) also adopts this bound.

---

[1]https://www.kaggle.com/datasets/ambityga/imagenet100/data
[2]PIL generally performs well for $\lambda \in [0.3, 0.9]$; see Appendix A.12 for details.

For the baselines, we follow the settings used in their original papers: EM, REM, TAP, NTGA, and SEP constrain perturbations under the $L_\infty$ norm with $\epsilon = 8/255$, while SP and AR constrain perturbations under the $L_2$ norm with $\epsilon = 1$. The original CUDA does not impose a fixed perturbation budget, for a fair comparison, we additionally evaluate a budget-matched ($L_\infty$ norm with $\epsilon = 8/255$) version of CUDA, and denote it as CUDA*.

## 4.1 Effectiveness of PIL Method on Different Datasets and Models

As shown in Table 1, we demonstrate the effectiveness of the proposed PIL method in generating unlearnable examples across four widely used datasets, which vary in image resolution and number of classes. The strong performance of PIL clearly indicates its ability to handle diverse image characteristics. Moreover, PIL generates unlearnable examples using only a linear model, without relying on specific architectural details such as convolutional layers in deep neural networks. Despite this, the resulting examples significantly degrade the performance of all tested DNNs, highlighting the architecture-independence of the method. These results collectively suggest that PIL is a promising approach for preventing unauthorized data exploitation.

Table 1: Test accuracies (%) on clean data for models trained on clean datasets ($\mathcal{D}_c$) or PIL-constructed unlearnable datasets ($\mathcal{D}_u$).

| Model | SVHN | | CIFAR-10 | | CIFAR-100 | | ImageNet Subset | |
|---|---|---|---|---|---|---|---|---|
| | $\mathcal{D}_c$ | $\mathcal{D}_u$ | $\mathcal{D}_c$ | $\mathcal{D}_u$ | $\mathcal{D}_c$ | $\mathcal{D}_u$ | $\mathcal{D}_c$ | $\mathcal{D}_u$ |
| ResNet-18 | 95.64 | **15.94** | 92.11 | **12.77** | 72.70 | **2.11** | 66.00 | **2.26** |
| ResNet-50 | 95.30 | **18.19** | 89.54 | **20.32** | 65.90 | **1.18** | 71.20 | **2.26** |
| VGG-19 | 95.22 | **9.12** | 90.61 | **15.22** | 64.57 | **1.40** | 36.04 | **1.36** |
| DenseNet-121 | 95.88 | **11.57** | 93.51 | **17.70** | 75.22 | **1.23** | 76.98 | **3.14** |
| MobileNet-V2 | 95.95 | **28.48** | 91.94 | **14.05** | 70.66 | **0.99** | 71.26 | **2.20** |

## 4.2 PIL Method Against Common Countermeasures

### 4.2.1 Data Augmentations.

**Common Augmentations.** In Table 1, we have applied the Basic augmentation, which includes horizontal flips and random crops. In this subsection, we further investigate the impact of representative data augmentation strategies on unlearnable examples. As summarized in Table 2, we evaluate nine augmentations. For simplicity, Perspective and ChannelShuffle are abbreviated as Persp and ChShuf in the table. Except for None, all augmentations are applied in combination with Basic. We highlight the top two performing methods under each augmentation strategy with a gray background. As shown in Table 2, PIL consistently demonstrates strong robustness across diverse data augmentations. It is worth noting that PIL achieves performance that is highly competitive with state-of-the-art methods like SEP across a wide range of augmentations. Furthermore, we compare the runtime efficiency of different methods in Table 5, where PIL is significantly more time-efficient than TAP, SEP, and AR.

Table 2: Clean test accuracies (%) on CIFAR-10 for different unlearnable examples under various data augmentation strategies. Closer to 10% is better. Gray background indicates top-2 methods.

| **Method** | None | Basic | Rotation | Persp | GrayScale | ChShuf | Cutout | CutMix | MixUp |
|---|---|---|---|---|---|---|---|---|---|
| Clean | 83.93 | 91.45 | 92.05 | 93.38 | 88.84 | 91.99 | 92.71 | 93.49 | 93.87 |
| EM | 21.43 | 24.82 | 28.67 | 29.77 | 89.15 | 57.64 | 23.47 | 29.39 | 48.26 |
| REM | 21.36 | 23.24 | 22.71 | 26.16 | 69.35 | 64.37 | 17.82 | 24.46 | 24.91 |
| TAP | 35.90 | 19.11 | 21.18 | 21.47 | 19.38 | 13.56 | 15.09 | 11.64 | 20.30 |
| SP | 16.06 | 23.92 | 23.67 | 25.28 | 82.79 | 73.04 | 23.74 | 21.11 | 26.41 |
| NTGA | 27.60 | 30.22 | 29.45 | 31.42 | 70.55 | 56.21 | 25.86 | 15.80 | 17.34 |
| CUDA* | 65.47 | 87.45 | 89.49 | 88.54 | 80.86 | 80.82 | 85.8 | 81.34 | 89.05 |
| SEP | 28.43 | 8.94 | 19.68 | 23.70 | 9.61 | 9.80 | 9.74 | 12.02 | 10.48 |
| AR | 16.89 | 17.57 | 11.31 | 13.40 | 38.38 | 14.25 | 10.57 | 14.05 | 15.87 |
| PIL (ours) | 14.70 | 12.87 | 18.15 | 19.30 | 17.01 | 10.88 | 14.62 | 10.79 | 11.05 |

**Note:** CUDA* uses the perturbation budget ($\epsilon = 8/255$), additional results can be found in Appendix. A.8.

**JPEG Compression.** Liu et al. (2023) has shown that JPEG compression is an effective counter-measure against unlearnable examples. Here, we investigate how different JPEG quality factors affect the defense capability against various unlearnable examples. Table 3 summarizes the clean test accuracies on CIFAR-10 under JPEG compression with quality factors ranging from 90 to 10. For each compression quality, the top two performing methods are highlighted with a gray background. As shown in the table, PIL and SP exhibit strong robustness under JPEG compression. Combined with our experiments in Section 5, which demonstrate that existing unlearnable examples consistently induce more linear behaviors in the trained DNNs, the superior performance of PIL and SP may suggest that constructing unlearnable examples based on linear separability is a more fundamental approach, leading to greater robustness.

Table 3: Clean test accuracies (%) on CIFAR-10 for different unlearnable examples under various JPEG compression qualities. Closer to 10% is better. Gray background indicates top-2 methods.

| Method | JPEG90 | JPEG80 | JPEG70 | JPEG60 | JPEG50 | JPEG40 | JPEG30 | JPEG20 | JPEG10 |
|---|---|---|---|---|---|---|---|---|---|
| Clean | 90.99 | 88.61 | 89.74 | 89.43 | 87.87 | 88.44 | 87.67 | 87.06 | 83.90 |
| EM | 25.75 | 34.04 | 44.15 | 51.58 | 55.12 | 61.13 | 70.94 | 72.50 | 80.94 |
| REM | 67.21 | 79.92 | 81.64 | 81.22 | 82.85 | 83.69 | 83.56 | 84.58 | 82.79 |
| TAP | 20.10 | 42.87 | 64.80 | 72.21 | 78.69 | 82.14 | 84.01 | 84.12 | 83.14 |
| NTGA | 41.57 | 52.14 | 53.70 | 55.97 | 63.48 | 62.24 | 67.31 | 72.97 | 74.64 |
| SEP | 12.18 | 49.66 | 64.09 | 76.46 | 83.08 | 85.99 | 87.08 | 86.21 | 82.43 |
| AR | 53.53 | 69.36 | 79.61 | 84.66 | 86.53 | 87.24 | 87.64 | 86.74 | 83.35 |
| SP | 26.17 | 31.74 | 29.94 | 33.1 | 34.72 | 40.50 | 40.20 | 54.04 | 79.34 |
| PIL (ours) | 35.26 | 36.97 | 41.55 | 43.64 | 50.87 | 52.05 | 58.37 | 67.89 | 76.71 |

### 4.2.2 ADVERSARIAL TRAINING.

To further evaluate the robustness of PIL, we conduct adversarial training (PGD-7) with varying perturbation budgets. As shown in Table 4, PIL consistently reduces test accuracy across all four $\epsilon$ settings compared to the clean baseline. Results for other unlearnable methods are provided in Appendix A.9. Although adversarial training is a strong countermeasure against nearly all unlearnable examples (Tao et al., 2021), it is highly time-consuming. Since unlearnable examples inevitably preserve visual features, there will always be ways for models to relearn from them. However, forcing an adversary to pay such a large computational cost—while still failing to fully recover accuracy—represents a practical victory for PIL, especially given its small time cost.

Table 4: **Adversarial Training.** CIFAR-10 test accuracy (%) under adversarial training with different perturbation budgets $\epsilon$. Closer to 10% is better.

| Method | $\epsilon = 2/255$ | $\epsilon = 4/255$ | $\epsilon = 8/255$ | $\epsilon = 16/255$ |
|---|---|---|---|---|
| Clean | 88.31 | 86.41 | 80.59 | 60.15 |
| PIL (ours) | **78.15** | **83.36** | **79.25** | **58.23** |

### 4.3 TIME COMPARISON

Table 5 presents the time required to generate unlearnable examples for various methods on the CIFAR-10 dataset. As shown, our PIL method is highly efficient: even compared with surrogate-free methods, it is only slightly slower than SP and CUDA\*, while being considerably faster than other surrogate-based approaches such as EM, SEP, TAP, and REM. Surrogate-free methods generally achieve higher speed because they avoid training and executing computationally intensive deep neural networks as surrogates. However, as shown in Section 4.2.1, PIL consistently outperforms SP and CUDA\* when evaluated under various data augmentation strategies, demonstrating that its efficiency does not come at the expense of robustness. Overall, these results indicate that PIL provides a practical and reliable approach for generating unlearnable examples, achieving both high efficiency and strong robustness across different experimental settings.

| Method | Time (s) |
|---|---|
| *Surrogate-based* | |
| PIL (ours) | 40.53 |
| EM | 1.65k |
| SEP | 26.78k |
| TAP | 40.14k |
| REM | 54.46k |
| *Surrogate-free* | |
| SP | 2.50 |
| CUDA\* | 9.11 |
| AR | 15.23k |

Table 5: Time comparison.

### 4.4 UNDERSTANDING PERTURBATION-DRIVEN LEARNING IN DNNS.

To investigate whether DNNs establish a clear correspondence between labels and perturbations, we evaluate them under four data configurations: (1) clean training set, (2) shuffled unlearnable training set $\mathcal{D}_s = \{(\boldsymbol{x}_i + \boldsymbol{\delta}_j, y_j)\}_{i=1}^n$, where each index $j$ is randomly sampled from $\{1, \ldots, n\}$, and (3) shuffled unlearnable test set, where clean test images are added with randomly selected perturbations from the training set. As shown in Table 6, models trained with some unlearnable methods, such as EM, SP, NTGA, and AR, still achieve relatively high accuracy when evaluated on shuffled sets, indicating that these models have learned to associate specific perturbations with corresponding labels. Our method (PIL) also achieves the high accuracies of 96.60% on the shuffled training set and 96.36% on the shuffled test set, demonstrating a strong perturbation-label correspondence.

Table 6: Accuracies (%) of ResNet-18 models trained with different unlearnable examples tested on clean, unshuffled and shuffled data.

| Testing \Training | EM | REM | TAP | SP | NTGA | SEP | AR | PIL (ours) |
|---|---|---|---|---|---|---|---|---|
| Clean Train | 23.87 | 20.02 | 19.90 | 23.93 | 31.33 | 10.69 | 11.13 | 22.42 |
| Shuffled Train | 44.41 | 17.71 | 17.19 | 94.10 | 88.96 | 27.12 | 99.52 | 96.60 |
| Shuffled Test | 44.26 | 17.97 | 17.21 | 93.64 | 88.49 | 26.71 | 99.52 | 96.36 |

## 5 PIL DO IMPROVE THE LINEARITY OF DNNS

In Section 3.2, we designed PIL with the explicit goal of forcing a deep neural network to learn a simple linear mapping. This leads to a natural question: does this "perturbation-induced linearization" actually occur in practice? More broadly, we hypothesize that inducing model linearity is a fundamental mechanism underlying the effectiveness of many unlearnable example methods, even those not explicitly designed for it. By forcing a high-capacity DNN to behave more like a low-capacity linear model, these perturbations effectively cripple its ability to learn complex, generalizable semantic features. In this section, we provide strong empirical evidence to support this Linearization Hypothesis.

**A Proxy for Linearity.** To quantify a model's degree of linearity, we draw inspiration from adversarial examples (Goodfellow et al., 2014). Goodfellow et al. demonstrated that the effectiveness of Fast Gradient Sign Method (FGSM) stems from the locally linear behavior of DNNs in high-dimensional space. A more linear model exhibits a flatter and more predictable local loss landscape, making it more vulnerable to FGSM. Accordingly, we use the performance drop under FGSM attacks as a proxy for linearity: a larger drop indicates stronger linear behavior.

**PIL Induces Strong Linear Behavior.** We first validate our hypothesis on PIL. We train ResNet-18 models on datasets containing a mixture of clean and PIL-perturbed examples. As shown in Table 7, the effect is striking. Even with only 10% of the data perturbed, the resulting model is noticeably more vulnerable to FGSM attacks across all attack strengths compared to a model trained on a similarly-sized clean subset. As the proportion of PIL-perturbed data increases, the model's performance under FGSM attack collapses dramatically, which confirms that PIL successfully induce a more linear behavior in the trained DNN, as intended by our design.

Table 7: Test accuracy and accuracy drop (%) under FGSM attack at different perturbation proportions, ResNet-18 models are trained with mixed clean and PIL perturbed data.

| FGSM Step | 10% | | 50% | | 90% | |
|---|---|---|---|---|---|---|
| | perturbed | clean | perturbed | clean | perturbed | clean |
| 0/255 | 92.33 (-0.0) | 90.83 (-0.0) | 89.73 (-0.0) | 87.69 (-0.0) | 74.81 (-0.0) | 68.78 (-0.0) |
| 1/255 | 61.93 (-30.4) | 61.84 (-28.99) | 45.33 (-44.4) | 59.69 (-28.0) | 19.6 (-55.21) | 48.56 (-20.22) |
| 2/255 | 38.13 (-54.2) | 40.23 (-50.6) | 22.13 (-67.6) | 38.23 (-49.46) | 7.24 (-67.57) | 32.64 (-36.14) |
| 4/255 | 17.39 (-74.94) | 23.79 (-67.04) | 9.11 (-80.62) | 21.29 (-66.4) | 3.05 (-71.76) | 15.16 (-53.62) |
| 8/255 | 7.95 (-84.38) | 14.51 (-76.32) | 5.84 (-83.89) | 13.6 (-74.09) | 2.61 (-72.2) | 6.34 (-62.44) |

**A General Phenomenon Across Methods.** To examine the generality of our hypothesis, we extend the analysis to a variety of existing unlearnable example methods (Appendix A.16) and provide a

snapshot in the main text for readability (Table 8). We report the *additional* accuracy drop under FGSM attacks when 50% of the training data is perturbed. The results are noteworthy: all tested methods, regardless of design, tend to render models more vulnerable to FGSM, as indicated by the fact that all elements in the table are greater than 0. This consistent trend suggests that inducing linearity is not unique to PIL, but rather a common mechanism of unlearnable examples.

Table 8: Additional accuracy drop (%) caused by different unlearnable examples under various FGSM attack steps at 50% perturbation.

| Step | EM | REM | TAP | SP | NTGA | SEP | AR | PIL (ours) |
|---|---|---|---|---|---|---|---|---|
| 1/255 | 13.37 | 17.48 | 25.44 | 14.26 | 17.50 | 9.60 | 25.15 | 16.40 |
| 2/255 | 13.95 | 21.15 | 24.51 | 16.27 | 21.10 | 11.48 | 18.39 | 18.14 |
| 4/255 | 9.08 | 21.92 | 13.24 | 12.20 | 16.00 | 7.94 | 7.12 | 14.22 |
| 8/255 | 4.65 | 18.76 | 1.38 | 7.67 | 10.50 | 1.20 | 3.04 | 9.80 |

This discovery offers a new perspective on what makes unlearnable examples effective. They appear to simplify the function that a DNN learns, constraining it to a lower-capacity, more linear regime. This insight explains why our linear-surrogate-based PIL method can achieve performance competitive with, or even superior to, methods that rely on computationally expensive deep surrogate models. While other methods may induce linearity as an indirect side effect of their complex optimization schemes, PIL targets this mechanism directly.

Therefore, effective data protection may not necessarily require complex adversarial machinery. Instead, directly inducing linearity offers a simpler and more efficient path. PIL provides a clear demonstration of this principle, delivering a practical and robust solution.

## 6 PARTIAL PERTURBATION: WHY DOESN'T ACCURACY DROP SIGNIFICANTLY?

In realistic scenarios, only a portion of the dataset may be perturbed. However, as shown in Figure 3, increasing the proportion of training samples replaced by PIL-perturbed examples does not cause a substantial drop in clean test accuracy. This naturally raises the question:

*Why does the model still maintain high accuracy even when a large fraction of the data is perturbed?*

This phenomenon is not unique to PIL. As reported by Liu et al. (2023), **all existing methods exhibit the same behavior**. Due to space constraints, we provide supporting results in Appendix A.10.1, Table 15.

A *qualitative* explanation, also noted in prior work (Huang et al., 2021), is that models can already achieve reasonably high accuracy with only a subset of clean samples, while perturbed samples contribute little to further improvement. This is confirmed by the red curve in Figure 3, which corresponds to models trained solely on clean data (resampled to maintain the same dataset size). Even with only 40% clean training data, the model achieves a clean test accuracy of 88.56%, and adding the remaining 60% perturbed samples provides negligible benefit.

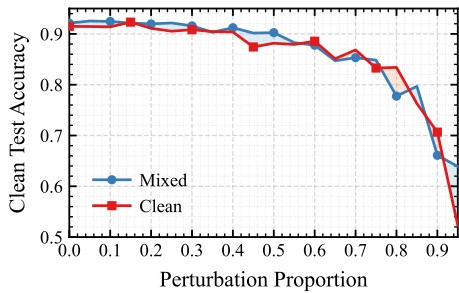

Figure 3: Clean test accuracy on CIFAR-10 under partial perturbation setting.

Our *quantitative* experiments further support this perspective. As shown in Appendix A.10.2, Figure 4, adding 90% new perturbed images (45k samples) yields an improvement equivalent to adding fewer than 10% new clean images (5k samples). Therefore, PIL effectively prevents DNNs from learning useful features from perturbed data.

However, PIL—like all existing methods—cannot prevent models from learning from the remaining clean samples. To better understand why unlearnable examples fail to stop this, we make the following Assumption 1, which is intuitive and can be empirically verified in Appendix A.4.1, Table 10:

**Assumption 1.** *Let $\mathcal{D}_c$ and $\mathcal{D}_u$ denote the clean and perturbed subsets of the training set, and $f_\theta$ denotes a model with parameter $\theta$. For all $\theta$ encountered during training:*

$$\mathbb{E}_{(\boldsymbol{x},y)\in\mathcal{D}_c}[\nabla_\theta\ell(f_\theta(\boldsymbol{x}),y)] \perp \mathbb{E}_{(\boldsymbol{x},y)\in\mathcal{D}_u}[\nabla_\theta\ell(f_\theta(\boldsymbol{x}),y)].$$

**Theorem 1.** *Let $\theta$ be the model parameters, and let $\alpha$ be the fraction of perturbed training data. Denote $L_c(\theta)$ and $L_u(\theta)$ as the average loss on the clean data $\mathcal{D}_c$ and the perturbed data $\mathcal{D}_u$, respectively. For a learning rate $\eta$, the change in the clean data loss after one gradient descent step is given by:*

$$L_c(\theta_{t+1}) - L_c(\theta_t) = -\eta\,\nabla_\theta L_c(\theta_t)\cdot[\alpha\nabla_\theta L_u(\theta_t) + (1-\alpha)\nabla_\theta L_c(\theta_t)]. \tag{9}$$

*Under **Assumption 1** (Gradient Orthogonality), i.e., $\nabla_\theta L_c(\theta_t)\cdot\nabla_\theta L_u(\theta_t) = 0$, the update simplifies to:*

$$L_c(\theta_{t+1}) - L_c(\theta_t) = -\eta(1-\alpha)\|\nabla_\theta L_c(\theta_t)\|^2.$$

The detailed analysis is in Appendix A.3.

Theorem 1 shows that unlearnable examples do not interfere with learning from clean data. In other words, they neither help nor hinder the learning process on clean samples. This decoupling can be explained by the observed gradient orthogonality between perturbed and clean samples. Since their gradients lie in nearly orthogonal directions, the perturbations do not alter the optimization for clean examples. Consequently, the model's generalization capability on clean data remains dependent on the actual clean samples available during training.

As the name suggests, unlearnable examples are designed to prevent the model learning useful features from the protected samples, but they do not inherently interfere with learning from clean ones. As illustrated in Fig. 3, the red and blue curves are closely intertwined, only plunging together when the perturbation ratio exceeds 80%. This pattern indicates that the sharp drop is not due to unlearnable examples suddenly becoming effective, but rather because the remaining clean samples in the dataset have become insufficient to sustain high model accuracy at that critical point. Thus, while partial perturbation does not cause the sharp accuracy drop seen under full perturbation, this outcome is an expected property of unlearnable examples.

Beyond the analyses presented in this section, we conducted additional experiments to further probe how PIL shapes models trained on PIL-perturbed data. (i) When only a single class is perturbed, the accuracy degradation is concentrated on that class, while the accuracy on other classes remains largely stable (Appendix A.11, Fig. 5). (ii) PIL generally enlarges the singular value spectrum of the input–output Jacobian, indicating amplified sensitivity along more directions (Appendix A.11, Fig. 6). (iii) The parameter gradients of samples within the same class become more similar after training on PIL-perturbed data, as evidenced by increased intra-class cosine similarity (Appendix A.4.2, Table 11). These complementary observations are consistent with the decoupling view: PIL suppresses learnable signal in the protected subset without disrupting learning from clean data, while also reshaping local geometry and class-specific behavior in a manner aligned with our main findings.

## 7 CONCLUSION

In this work, we propose Perturbation-Induced Linearity (PIL), a method that enables the generation of unlearnable examples using only linear models, offering a simple yet effective approach with low computational cost. Through extensive experiments, we demonstrate that PIL remains effective under various data augmentation strategies and adversarial training. Beyond empirical evaluation, we provide the first theoretical analysis of the limitation of unlearnable examples under partial perturbation settings. More importantly, we reveal a fundamental insight: unlearnable perturbations induce stronger linear behavior in deep models, thereby impairing their ability to learn meaningful representations. We believe this insight opens new directions for designing stronger data protection mechanisms.

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

## A  APPENDIX

### A.1  EVALUATION CRITERIA FOR UNLEARNABILITY

There is currently no unified evaluation criterion for "unlearnability." Existing studies adopt different perspectives, which can be broadly categorized into two groups, as summarized in Table 9. We note that the *Minimal Accuracy* criterion may not fully capture the learning behavior of models, because low accuracy does not necessarily mean that no features have been acquired. For example, in (Chen et al., 2022), the model maps one class to another during prediction.

We prefer the *Random Guessing* criterion; more specifically, we define unlearnability as preventing the model from extracting useful features from the protected data, rather than deliberately degrading its performance. Specifically:

- If all training samples are protected, the model should behave as if trained on random guesses, i.e., no effective learning occurs.

- If 30% of the samples are clean and 70% are protected, the model should perform as though it was trained only on the 30% clean samples, meaning that the protected data **does not contribute** to learning.

Table 9: Evaluation Criteria for Unlearnability in Prior Work

| Category | Brief Criterion | Representative Works |
|---|---|---|
| **Random Guessing** | Models' test accuracy closer to random guessing is better | Huang et al. (2021), Yu et al. (2022) Fu et al. (2022), Wu et al. (2022) |
| **Minimal Accuracy** | Minimizes models' test accuracy as low as possible | Fowl et al. (2021), Yuan & Wu (2021), Sandoval-Segura et al. (2022), Chen et al. (2022), Feng et al. (2019), Sandoval-Segura et al. (2023), Shen et al. (2019), van Vlijmen et al. (2022) |

### A.2  FUNCTION DEFINITIONS

We briefly define the key functions used in our formulation:

- **Cross-entropy loss.** Denoted as $L_{\mathrm{CE}}(\boldsymbol{p}, \boldsymbol{y})$, where $\boldsymbol{p} = \mathrm{softmax}(\boldsymbol{z})$ is the predicted probability vector and $\boldsymbol{y} = \mathrm{onehot}(y)$ is the ground-truth label. This loss measures dissimilarity between predicted and true distributions:

$$L_{\mathrm{CE}}(\boldsymbol{p}, \boldsymbol{y}) = -\sum_{i=1}^{K} y_i \log p_i \tag{10}$$

- **KL divergence.** The Kullback–Leibler divergence $L_{\mathrm{KL}}(\boldsymbol{p}, \boldsymbol{q})$ quantifies how the distribution $\boldsymbol{p}$ diverges from reference distribution $\boldsymbol{q}$:

$$L_{\mathrm{KL}}(\boldsymbol{p}, \boldsymbol{q}) = \sum_{i=1}^{K} p_i \log \frac{p_i}{q_i} \tag{11}$$

### A.3 THEORETICAL ANALYSIS ON THE PROPERTY OF PARTIAL PERTURBATION IN UNLEARNABLE ATTACKS

**Problem Setup.** Let the training set $\mathcal{D}$ contain $n$ samples, with a fraction $\alpha$ ($0 < \alpha < 1$) of them perturbed by an unlearnable example methods, and the remaining $(1 - \alpha)n$ samples left clean. Denote:

- Clean sample loss: $L_c(\theta) = \frac{1}{(1-\alpha)n} \sum_{(\boldsymbol{x}_i, y) \in \mathcal{D}_c} \ell(f_\theta(\boldsymbol{x}_i), y)$,
- Perturbed sample loss: $L_u(\theta) = \frac{1}{\alpha n} \sum_{(\boldsymbol{x}_i, y) \in \mathcal{D}_u} \ell(f_\theta(\boldsymbol{x}_i + \boldsymbol{\delta}_i), y)$

where $\mathcal{D}_c$ and $\mathcal{D}_u$ represent the sets of clean and perturbed samples respectively, and $\theta$ denotes the model parameters.

**Total Gradient Decomposition.** The overall training loss is a weighted sum of the two losses:

$$L(\theta) = \alpha L_u(\theta) + (1 - \alpha)L_c(\theta), \tag{12}$$

so the total gradient becomes:

$$\nabla_\theta L(\theta) = \alpha \nabla_\theta L_u(\theta) + (1 - \alpha)\nabla_\theta L_c(\theta). \tag{13}$$

**Parameter Update and Learning Signal.** Using gradient descent with learning rate $\eta$, the update rule is:

$$\theta_{t+1} = \theta_t - \eta \nabla_\theta L(\theta). \tag{14}$$

The change in loss on the clean data subset is (assuming $\eta$ is sufficiently small):

$$\begin{aligned} \Delta L_c &= L_c(\theta_{t+1}) - L_c(\theta_t) \\ &= \nabla_\theta L_c(\theta_t) \cdot (\theta_{t+1} - \theta_t). \end{aligned} \tag{15}$$

Substituting the update rule (Eq. 14),

$$\begin{aligned} \Delta L_c &= -\eta \nabla_\theta L_c(\theta_t) \cdot \nabla_\theta L(\theta_t) \\ &= -\eta[\alpha \nabla_\theta L_c(\theta_t) \cdot \nabla_\theta Lu(\theta_t) + (1 - \alpha)\nabla_\theta L_c(\theta_t) \cdot \nabla_\theta Lc(\theta_t)]. \end{aligned} \tag{16}$$

Thus,

$$L_c(\theta_{t+1}) - L_c(\theta_t) = -\eta \, \nabla_\theta L_c(\theta_t) \cdot [\alpha \nabla_\theta L_u(\theta_t) + (1 - \alpha)\nabla_\theta L_c(\theta_t)] . \tag{17}$$

Then, under Assumption 1, the expected gradients of the clean and perturbed subsets are orthogonal, so the update reduces to

$$L_c(\theta_{t+1}) - L_c(\theta_t) = -\eta(1 - \alpha)\|\nabla_\theta L_c(\theta_t)\|^2. \tag{18}$$

**Conclusion.** The update on the clean loss behaves as if gradient descent were performed solely on clean samples, with the learning rate scaled by a factor of $(1 - \alpha)$.

### A.4 GRADIENT ANALYSIS BETWEEN UNLEARNABLE AND CLEAN SAMPLES

#### A.4.1 EMPIRICAL VERIFICATION OF GRADIENT ORTHOGONALITY

Intuitively, if we want a deep neural network to fail to learn meaningful features, the gradients produced by the perturbed samples should be approximately orthogonal to those from the clean samples.

To justify Assumption 1, we empirically examine the alignment between gradients produced by clean and perturbed samples. Concretely, given a clean image-label pair $(\boldsymbol{x}, y)$ and its perturbed counterpart $(\boldsymbol{x} + \boldsymbol{\delta}, y)$, we define the gradients of the training loss with respect to model parameters as

$$g_{\text{clean}} = \nabla_\theta \ell(f_\theta(\boldsymbol{x}), y), \quad g_{\text{pert}} = \nabla_\theta \ell(f_\theta(\boldsymbol{x} + \boldsymbol{\delta}), y).$$

In practice, we compute gradient similarity at the batch level. Given a mini-batch $\mathcal{B} = \{(\boldsymbol{x}_j, y_j)\}_{j=1}^{m}$, we define the averaged gradients as

$$\bar{g}_{\text{clean}} = \frac{1}{m}\sum_{j=1}^{m}\nabla_\theta \ell(f_\theta(\boldsymbol{x}_j), y_j), \quad \bar{g}_{\text{pert}} = \frac{1}{m}\sum_{j=1}^{m}\nabla_\theta \ell(f_\theta(\boldsymbol{x}_j + \boldsymbol{\delta}_j), y_j).$$

The cosine similarity is then computed as

$$\cos(\bar{g}_{\text{clean}}, \bar{g}_{\text{pert}}) = \frac{\langle \bar{g}_{\text{clean}}, \bar{g}_{\text{pert}}\rangle}{\|\bar{g}_{\text{clean}}\| \cdot \|\bar{g}_{\text{pert}}\|}.$$

We evaluate multiple unlearnable example methods on CIFAR-10 with ResNet-18. As shown in Table 10, across all methods the gradients of perturbed samples are nearly orthogonal to those of clean samples, within only 10 epochs of training on unlearnable datasets.

Our experimental results align with the intuition that the gradients produced by the perturbed samples should be approximately orthogonal to those from the clean samples. Moreover, the Gradient Orthogonality Assumption provides a reasonable and empirically validated foundation for Theorem 1.

Table 10: Average cosine similarity between the batch-averaged gradients of clean samples ($\bar{g}_c$) and protected samples ($\bar{g}_u$). The similarity for each batch is computed as $\cos(\bar{g}_{\text{clean}}, \bar{g}_{\text{pert}}) = \frac{\langle \bar{g}_{\text{clean}}, \bar{g}_{\text{pert}}\rangle}{\|\bar{g}_{\text{clean}}\| \cdot \|\bar{g}_{\text{pert}}\|}$, and then averaged over an epoch.

| Epoch | EM | REM | TAP | SP | NTGA | SEP | AR | PIL (ours) |
|---|---|---|---|---|---|---|---|---|
| 0 (Init) | 0.649 | 0.694 | 0.623 | 0.695 | 0.669 | 0.630 | 0.725 | 0.634 |
| 1 | -0.117 | 0.997 | 1.000 | 0.205 | 0.991 | 1.000 | 1.000 | 0.561 |
| 2 | -0.000 | 0.869 | 0.999 | -0.090 | 0.425 | 0.999 | 1.000 | 0.276 |
| 3 | 0.234 | -0.193 | 0.936 | 0.066 | 0.607 | 0.845 | 0.999 | 0.030 |
| 4 | 0.125 | 0.053 | 0.055 | -0.014 | 0.226 | 0.740 | 0.971 | 0.120 |
| 5 | 0.040 | 0.348 | 0.395 | 0.002 | -0.075 | 0.330 | 0.341 | 0.054 |
| 6 | -0.044 | -0.115 | 0.274 | -0.066 | -0.025 | 0.484 | 0.471 | 0.066 |
| 7 | -0.056 | -0.069 | -0.166 | -0.074 | 0.051 | -0.228 | 0.440 | -0.043 |
| 8 | -0.001 | -0.021 | 0.113 | -0.083 | -0.019 | 0.113 | 0.381 | -0.086 |
| 9 | 0.002 | -0.002 | -0.002 | -0.090 | -0.050 | -0.039 | 0.193 | -0.084 |
| 10 | -0.001 | 0.006 | -0.017 | -0.093 | -0.062 | -0.047 | -0.039 | -0.082 |

### A.4.2 INTRA-CLASS GRADIENT SIMILARITY ANALYSIS

In the previous subsection, we discussed that the overall gradients between clean and perturbed samples are nearly orthogonal. Here, we further investigate how unlearnable examples affect the gradients of samples within the same class: do they make the gradients more similar or more diverse?

Table 11 summarizes the average cosine similarity between per-sample loss gradients with respect to model parameters for all sample pairs within each class, computed using the clean training set on models trained either on clean data or on unlearnable datasets. For the clean model, intra-class gradient similarity is relatively low (typically in the interval $[0.1, 0.2]$), indicating that gradients of samples within the same class retain substantial diversity.

In contrast, models trained on perturbed datasets consistently exhibit higher intra-class gradient similarity, commonly in the interval $[0.3, 0.7]$. This demonstrates that unlearnable perturbations interfere with the training process, causing the trained model to produce more aligned gradients for samples of the same class. Higher intra-class gradient similarity reflects a collapse in the gradient directions across samples, which implies that the model captures less diverse information from individual samples. These results provide quantitative evidence that unlearnable examples hinder the learning of diverse features, effectively reducing the model's ability to represent intra-class variation.

### A.5 DETAILED EXPERIMENTAL SETTING FOR PIL

We initialize perturbations $\{\boldsymbol{\delta}_i\}_{i=1}^{n}$ from a uniform distribution $\text{Uniform}(-\epsilon, \epsilon)$ with budget $\epsilon = 8/255$. Updates use step size $\alpha = 8/2550$, and the balancing factor $\lambda = 0.9$ to balance the objectives

Table 11: Intra-class gradient cosine similarity for each method across CIFAR-10 classes.

| Method | 0 | 1 | 2 | 3 | 4 | 5 | 6 | 7 | 8 | 9 |
|--------|-----|-----|-----|-----|-----|-----|-----|-----|-----|-----|
| Clean | 0.121 | 0.181 | 0.107 | 0.163 | 0.099 | 0.147 | 0.156 | 0.162 | 0.187 | 0.199 |
| EM | 0.266 | 0.273 | 0.283 | 0.303 | 0.350 | 0.305 | 0.406 | 0.335 | 0.348 | 0.325 |
| REM | 0.320 | 0.528 | 0.345 | 0.393 | 0.365 | 0.381 | 0.510 | 0.472 | 0.568 | 0.477 |
| TAP | 0.349 | 0.533 | 0.335 | 0.377 | 0.349 | 0.316 | 0.397 | 0.372 | 0.291 | 0.397 |
| SP | 0.432 | 0.413 | 0.368 | 0.418 | 0.332 | 0.387 | 0.393 | 0.370 | 0.461 | 0.477 |
| NTGA | 0.356 | 0.281 | 0.287 | 0.373 | 0.283 | 0.286 | 0.334 | 0.389 | 0.303 | 0.331 |
| SEP | 0.371 | 0.473 | 0.358 | 0.486 | 0.286 | 0.602 | 0.389 | 0.466 | 0.445 | 0.415 |
| AR | 0.727 | 0.612 | 0.716 | 0.611 | 0.701 | 0.728 | 0.713 | 0.727 | 0.688 | 0.554 |
| PIL (ours) | 0.431 | 0.396 | 0.356 | 0.386 | 0.332 | 0.400 | 0.307 | 0.336 | 0.387 | 0.328 |

in Eq. 3 and Eq. 4. The linear model $f_{\text{lin}}(\cdot; \boldsymbol{w})$ is optimized for $M = 30$ iterations, with perturbations updated for $N = 30$ steps, using a cosine annealing learning rate schedule. For SVHN, CIFAR-10, and CIFAR-100, we use an initial learning rate $\eta = 0.003$ and momentum 0.9. For the ImageNet Subset, we use $\eta = 0.03$, weight decay $1 \times 10^{-4}$, momentum 0.9, and apply gradient clipping with threshold 1.0.

## A.6 DETAILS OF BASELINE METHODS

- **Error-Minimizing (EM)** (Huang et al., 2021): EM generates unlearnable examples by minimizing the classification error of perturbed images using a surrogate deep neural network. The optimization alternates between updating the perturbations and training the surrogate model. We use the official implementation provided by the authors.[3]

- **Robust Error-Minimizing (REM)** (Fu et al., 2022): REM replaces the normally-trained surrogate in EM with an adversarially-trained model, making the attacks more robust against adversarial training. We use the official implementation provided by the authors.[4]

- **Targeted Adversarial Poisoning (TAP)** (Fowl et al., 2021): TAP uses a fixed, pretrained surrogate model and minimizes the classification loss with respect to target labels rather than the original labels. We use the official implementation provided by the authors.[5]

- **Synthetic Perturbations (SP)** (Yu et al., 2022): SP generates perturbations by sampling linearly separable Gaussian samples, which are then up-scaled to match the input image size. Perturbations sampled from the same Gaussian distribution are assigned to the same class. We use the official implementation provided by the authors.[6]

- **Neural Tangent Generalization Attacks (NTGA)** (Yuan & Wu, 2021): NTGA uses the neural tangent kernel (Jacot et al., 2018) as the surrogate to model the training dynamics of a class of wide DNNs and then leverages it to generate perturbations. We use the perturbed datasets provided by the authors.[7]

- **Self-Ensemble Protection (SEP)** (Chen et al., 2022): SEP generates perturbations by ensembling intermediate checkpoints obtained during training on the clean dataset. We use the official implementation provided by the authors.[8]

- **AutoRegressive poisoning (AR)** (Sandoval-Segura et al., 2022): AR leverages an autoregressive process to generate perturbations that are favored by CNNs during training. We use the official implementation provided by the authors.[9]

- **Convolution-based Unlearnable Datasets (CUDA)** (Sadasivan et al., 2023): CUDA constructs unlearnable examples by applying class-wise convolutions with randomly generated filters, encour-

---

[3]https://github.com/HanxunH/Unlearnable-Examples/

[4]https://github.com/fshp971/robust-unlearnable-examples/

[5]https://github.com/lhfowl/adversarial_poisons

[6]https://github.com/dayu11/Availability-Attacks-Create-Shortcuts/

[7]https://github.com/lionelmessi6410/ntga

[8]https://github.com/Sizhe-Chen/SEP

[9]https://github.com/psandovalsegura/autoregressive-poisoning

aging the model to associate filters with labels rather than to learn meaningful features from the clean data. We use the official implementation released by the authors.[10]

## A.7 EFFECT OF PRETRAINING ON PIL

We evaluate the performance of PIL with and without pretraining under various data augmentations. The average test accuracy of models trained on the unlearnable dataset is reported in Table 12.

Table 12: Test accuracy (%) of models trained on PIL-protected datasets under different augmentations. Lower is better. PIL-pre means PIL with pretraining; PIL-np means PIL without pretraining.

| Method | None | Basic | Rotation | Persp | Gray | ChShuf | Cutout | CutMix | MixUp |
|---|---|---|---|---|---|---|---|---|---|
| PIL-np | 15.94 | 13.24 | 27.99 | 23.70 | 24.39 | 34.13 | **10.90** | 12.72 | 29.36 |
| PIL-pre | **14.70** | **12.87** | **18.15** | **19.30** | **17.01** | **10.88** | 14.62 | **10.79** | **11.05** |

PIL without pretraining still effectively degrades model performance, especially under strong augmentations such as Cutout and CutMix. However, pretraining leads to significantly stronger protection, particularly for augmentations like Rotation, ChannelShuffle, and MixUp. These results demonstrate that while pretraining is not strictly necessary for PIL, it provides benefits in scenarios with heavy data augmentation.

## A.8 ADDITIONAL DATA AUGMENTATIONS RESULTS FOR SECTION 4.2.1

In this section, we report the results of the original CUDA method as a supplement to the main text. It is important to note that the original version of CUDA relaxes the imperceptibility constraint: on CIFAR-10, its average per-pixel perturbation reaches approximately $21.87/255$, which is substantially larger than the perturbation budgets used by other baselines (typically $L_\infty = 8/255$).

For a fair comparison, we include only the clipped variant of CUDA (denoted as CUDA*), which is constrained to the standard $L_\infty = 8/255$ budget, in the main tables. The original CUDA is shown here solely for completeness. As reported in Table 13, once the perturbation is clipped to $8/255$, the protection effect of CUDA* becomes noticeably weaker.

Table 13: Clean test accuracies (%) on CIFAR-10 for different unlearnable examples under various data augmentation strategies. Closer to 10% is better. Gray background indicates top-2 methods.

| Method | None | Basic | Rotation | Persp | GrayScale | ChShuf | Cutout | CutMix | MixUp |
|---|---|---|---|---|---|---|---|---|---|
| Clean | 83.93 | 91.45 | 92.05 | 93.38 | 88.84 | 91.99 | 92.71 | 93.49 | 93.87 |
| EM | 21.43 | 24.82 | 28.67 | 29.77 | 89.15 | 57.64 | 23.47 | 29.39 | 48.26 |
| REM | 21.36 | 23.24 | 22.71 | 26.16 | 69.35 | 64.37 | 17.82 | 24.46 | 24.91 |
| TAP | 35.90 | 19.11 | 21.18 | 21.47 | 19.38 | 13.56 | 15.09 | 11.64 | 20.30 |
| SP | 16.06 | 23.92 | 23.67 | 25.28 | 82.79 | 73.04 | 23.74 | 21.11 | 26.41 |
| NTGA | 27.60 | 30.22 | 29.45 | 31.42 | 70.55 | 56.21 | 25.86 | 15.80 | 17.34 |
| CUDA | 10.35 | 24.74 | 29.53 | 30.70 | 23.74 | 25.72 | 21.62 | 23.95 | 23.54 |
| CUDA* | 65.47 | 87.45 | 89.49 | 88.54 | 80.86 | 80.82 | 85.8 | 81.34 | 89.05 |
| SEP | 28.43 | 8.94 | 19.68 | 23.70 | 9.61 | 9.80 | 9.74 | 12.02 | 10.48 |
| AR | 16.89 | 17.57 | 11.31 | 13.40 | 38.38 | 14.25 | 10.57 | 14.05 | 15.87 |
| PIL (ours) | 14.70 | 12.87 | 18.15 | 19.30 | 17.01 | 10.88 | 14.62 | 10.79 | 11.05 |

**Note:** CUDA* uses the perturbation budget ($\epsilon = 8/255$), whereas CUDA uses its original larger budget.

## A.9 ADDITIONAL ADVERSARIAL TRAINING RESULTS FOR SECTION 4.2.2

As shown in Table 14, REM exhibits strong resistance under adversarial training with smaller perturbation budgets (e.g., $\epsilon \leq 4/255$). However, under large perturbation budgets, PIL outperforms REM. Apart from REM, all other methods demonstrate relatively similar performance under adversarial training. We believe that the performance of our PIL method can be further enhanced by introducing

---

[10]https://github.com/vinusankars/Convolution-based-Unlearnability

mechanisms similar to REM, which generates perturbations jointly with adversarial training. We leave the exploration of this direction for future work.

All adversarial training experiments were conducted using PGD-7 with an attack step size of $\epsilon/4$ for each perturbation budget.

Table 14: **Adversarial Training.** CIFAR-10 test accuracy (%) under adversarial training with different perturbation budgets $\epsilon$. Closer to 10% is better.

| Method | $\epsilon = 2/255$ | $\epsilon = 4/255$ | $\epsilon = 8/255$ | $\epsilon = 16/255$ |
|---|---|---|---|---|
| Clean | 88.31 | 86.41 | 80.59 | 60.15 |
| EM | 87.04 | 85.91 | 80.21 | 62.53 |
| REM | 29.01 | 42.84 | 82.67 | 62.88 |
| TAP | 80.56 | 83.93 | 77.77 | 60.01 |
| SP | 69.65 | 84.70 | 79.79 | 59.65 |
| NTGA | 79.46 | 82.52 | 77.53 | 57.68 |
| SEP | 80.63 | 74.34 | 72.17 | 63.22 |
| AR | 71.07 | 73.11 | 71.72 | 64.29 |
| PIL (ours) | 78.15 | 83.36 | 79.25 | 58.23 |

## A.10 ADDITIONAL RESULTS FOR SECTION 6

### A.10.1 PARTIAL PERTURBATION FOR OTHER METHODS

In this section, we conduct a detailed study on how various unlearnable examples perform when only a portion of the training data is perturbed. As shown in Table 15, the clean test accuracy consistently drops as the perturbation portion increases. However, even when up to 90% of the training data is perturbed, the degradation in accuracy still falls short compared to the fully perturbed setting.

We provide a detailed analysis of this phenomenon for our PIL method in Section 6, where we explore potential reasons for its limited effectiveness under partial perturbation. These insights may also be applicable to other methods.

Table 15: Clean test accuracy (%) on CIFAR-10 for different unlearnable example generation methods when only a portion of the training set is perturbed.

| Method | 0.1 | 0.2 | 0.3 | 0.4 | 0.5 | 0.6 | 0.7 | 0.8 | 0.9 |
|---|---|---|---|---|---|---|---|---|---|
| EM | 92.70 | 91.95 | 88.72 | 89.41 | 89.45 | 87.07 | 82.99 | 82.15 | 70.37 |
| REM | 91.89 | 91.99 | 91.76 | 89.14 | 89.03 | 83.13 | 86.93 | 79.80 | 74.86 |
| TAP | 91.55 | 91.41 | 88.81 | 91.03 | 89.79 | 88.19 | 87.35 | 83.92 | 80.40 |
| SP | 92.66 | 91.88 | 92.33 | 89.43 | 88.74 | 85.88 | 85.80 | 81.62 | 74.30 |
| NTGA | 91.48 | 91.83 | 91.44 | 89.88 | 89.61 | 89.10 | 85.00 | 81.13 | 71.49 |
| SEP | 91.53 | 91.18 | 91.05 | 90.99 | 90.01 | 89.25 | 86.99 | 85.70 | 82.47 |
| AR | 92.96 | 91.91 | 92.37 | 91.13 | 90.62 | 87.80 | 87.39 | 84.96 | 79.55 |
| PIL (ours) | 92.95 | 91.18 | 91.20 | 90.11 | 89.34 | 89.71 | 83.72 | 82.26 | 76.97 |

### A.10.2 PERTURBED DATA CONTRIBUTES LITTLE ACCURACY INCREASE

As qualitatively discussed in Section 6, perturbed samples contribute very little to the accuracy improvement of models. Here, we provide a quantitative analysis to support this observation.

As shown in Figure 4, we investigate how the model trained on a mix of $\eta$ portion clean and $1 - \eta$ portion perturbed data compares to models trained on only clean data. Specifically, we estimate how much additional clean data would be needed to achieve the same accuracy as that provided by the perturbed portion. The results show that across all settings where $\eta \in \{0.1, 0.2, \ldots, 0.8\}$, the accuracy benefit contributed by the $1 - \eta$ portion of perturbed data is equivalent to using no more than an additional 20% of clean data. This result clearly indicates that the perturbations introduced by PIL effectively hinder the model from learning meaningful representations from the perturbed data.

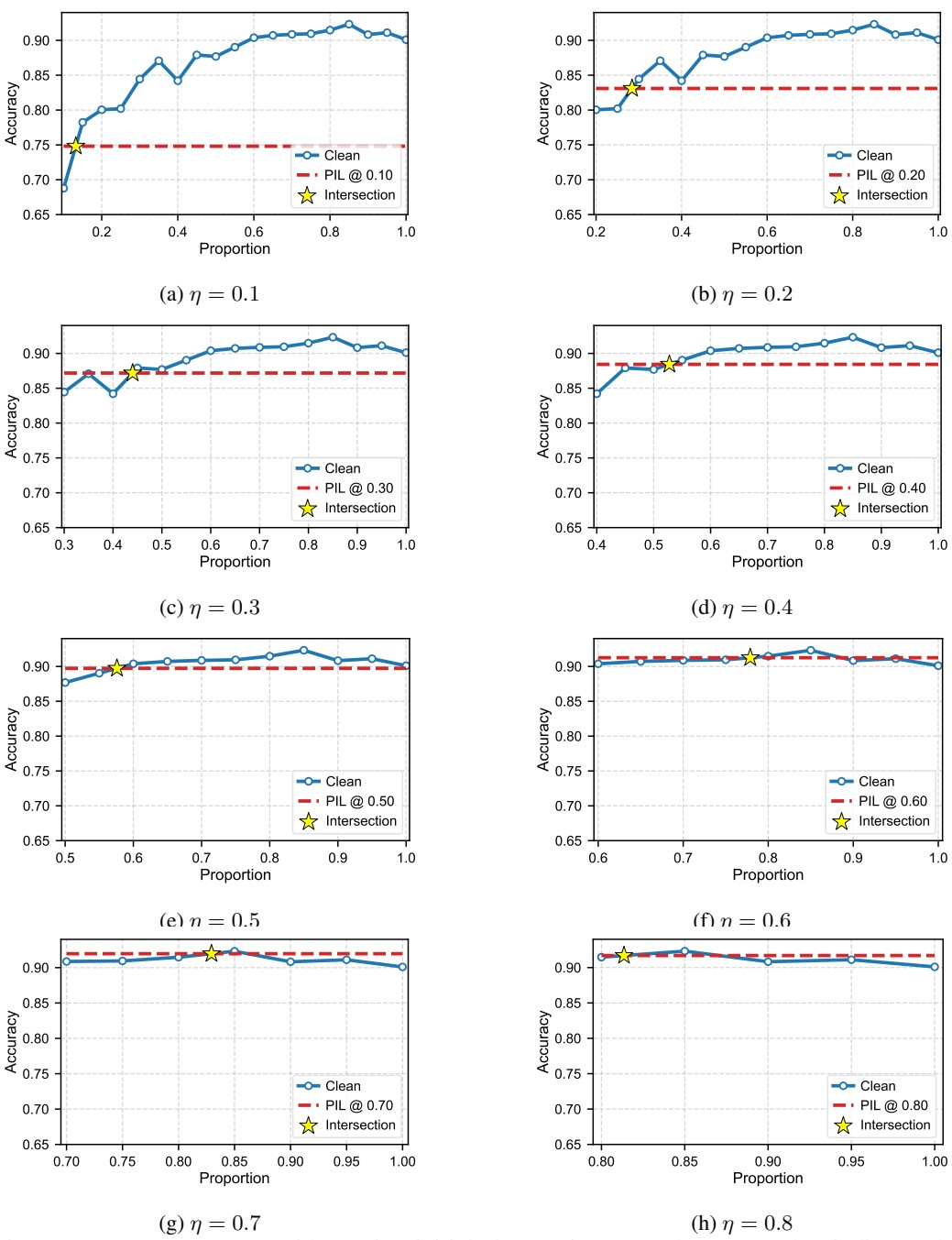

Figure 4: Accuracy curves with varying initial clean ratios ($\eta$). The star marker indicates the intersection point where the Clean curve crosses the PIL baseline. The x-axis (proportion) represents the total fraction of clean training data used relative to the entire dataset. (a)-(f) demonstrate the progressive impact of increasing $\eta$ from 0.1 to 0.8.

A.11 PERTURBATION PREVENTS SEMANTIC LEARNING

In this section, we provide further experimental evidence that unlearnable perturbations prevent models from acquiring meaningful semantic features.

Figures 5a and 5b show the model's behavior when all classes in the training set are perturbed. In this setting, the model exhibits high uncertainty across all test samples, as indicated by elevated entropy values, and tends to misclassify them into a few dominant categories such as "Car", "Bird", and "Truck". The corresponding confusion matrix confirms this collapse in classification diversity.

In contrast, Figures 5c and 5d display results when only the "Bird" class is perturbed. Here, the model maintains low prediction entropy and high accuracy for the unperturbed classes, but fails to learn useful representations for the "Bird" class, showing high entropy and almost random predictions for those samples.

These findings support our hypothesis: unlearnable perturbations effectively block learning of semantic information for perturbed classes, without significantly affecting the learning of clean ones.

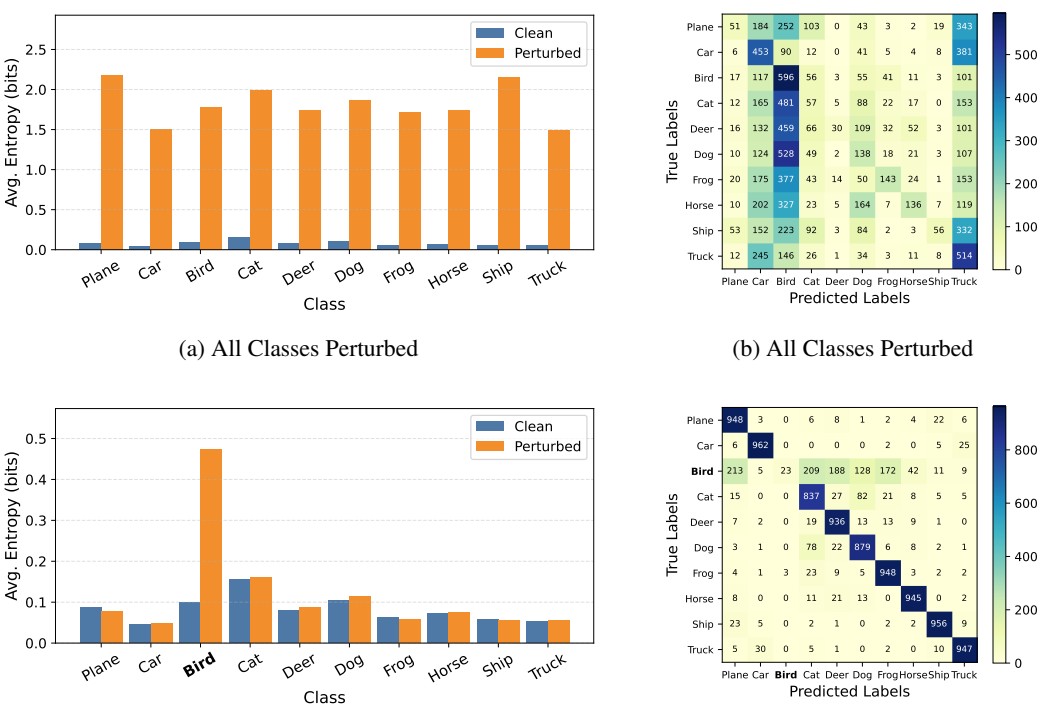

(a) All Classes Perturbed                    (b) All Classes Perturbed

(c) Class "Bird" Perturbed                    (d) Class "Bird" Perturbed

Figure 5: Effect of perturbations on prediction uncertainty and class confusion. Subfigures (a) and (c) show entropy of predictions on the clean test set, where higher values indicate greater uncertainty. Subfigures (b) and (d) present the corresponding confusion matrices.

To further understand the effect of PIL on the model's behavior, we also measured the Jacobian singular value spectra.

We computed the Jacobian singular value spectra on the clean CIFAR-10 test set for two ResNet-18 models: one trained on the clean CIFAR-10 training set, and the other trained on the PIL-perturbed CIFAR-10 training set. For each of the 10,000 test images, we calculated the singular values of the Jacobian (sorted in descending order), and then averaged the resulting singular value spectra across all test samples (Figure 6).

We observe that the singular values increase across almost all modes after applying PIL. In particular, the largest singular value rises from 69.03 to 72.76, and the smallest from 0.334 to 0.799. This overall elevation indicates that the model's output is more sensitive to input perturbations along multiple directions, suggesting that PIL increases the uncertainty in class selection and distributes the sensitivity more evenly across the input space.

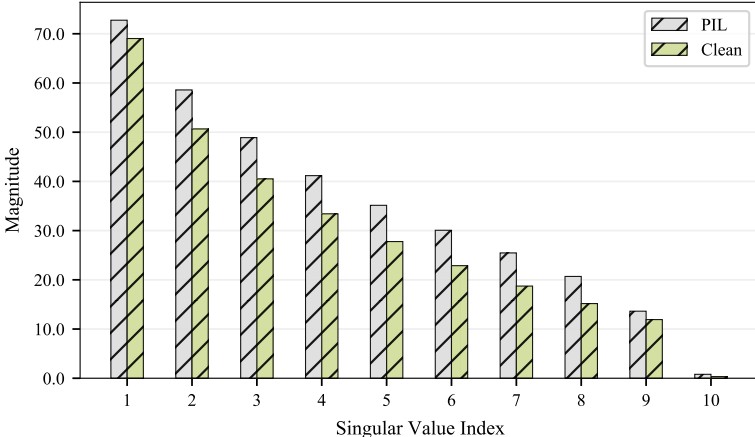

Figure 6: Average Jacobian singular value spectra on the CIFAR-10 test set for ResNet-18 models trained on clean and PIL-perturbed training data.

## A.12 PARAMETER SELECTION

We sweep the balancing factor $\lambda$ and evaluate PIL with various data augmentation, including two ablation cases at $\lambda = 0$ and $\lambda = 1$. When $\lambda \in (0, 1)$, both Eq. 3 and Eq. 4 are jointly optimized, whereas $\lambda = 0$ and $\lambda = 1$ correspond to using only Eq. 3 or only Eq. 4, respectively. As shown in Table 16, PIL consistently achieves low test accuracy across a broad interval of $\lambda \in [0.3, 0.9]$, indicating that the method is not sensitive to this hyperparameter. Notably, the most stable performance across all augmentation settings appears in $\lambda = 0.9$, where the perturbed datasets consistently reduce accuracy to below 20%.

The ablation settings further clarify the contribution of each loss component. When $\lambda = 1$ (removing Eq. 3), PIL still provides strong protection, although the performance is slightly less stable across augmentations compared to the best mixed settings ($\lambda = 0.9$). In contrast, when $\lambda = 0$ (removing Eq. 4), PIL exhibits almost no protective effect, which aligns with our expectation that Eq. 4 is the primary driver of unlearnability.

We now discuss the role of Eq. 3 in more detail. During the design of our method, we noted that even a purely linear classifier can achieve around 40% accuracy on CIFAR-10. Motivated by this, Eq. 3 was introduced to suppress the true features that a linear model would otherwise rely on, ensuring that models trained on the PIL-generated unlearnable dataset focus primarily on the perturbation–label correspondence rather than meaningful class information. In practice, as shown in Table 16, jointly using both loss terms leads to stronger and more stable protection across diverse settings. Especially our final choice, $\lambda = 0.9$, provides both strong and stable performance.

Table 16: CIFAR-10 test accuracy (%) of PIL with varying balancing factor $\lambda$ under different augmentation strategies. Closer to 10% is better.

| **Augs** | 0.0 | 0.1 | 0.2 | 0.3 | 0.4 | 0.5 | 0.6 | 0.7 | 0.8 | 0.9 | 1.0 |
|---|---|---|---|---|---|---|---|---|---|---|---|
| None | 82.97 | 31.30 | 13.16 | 7.20 | 7.64 | **10.60** | 13.98 | 14.96 | 11.75 | 17.08 | 18.76 |
| Basic | 90.71 | 83.99 | 56.72 | 38.57 | 28.46 | 26.13 | 20.68 | 21.78 | 27.16 | 15.98 | **13.12** |
| Rotation | 88.56 | 78.69 | 53.11 | 35.45 | 24.40 | **17.38** | 21.43 | 17.79 | 22.53 | 17.41 | 20.10 |
| Cutout | 92.54 | 79.09 | 52.46 | 28.26 | 23.82 | 17.64 | 13.10 | 18.25 | 22.94 | **12.34** | 21.58 |
| CutMix | 92.20 | 78.66 | 32.05 | 27.41 | 17.59 | 14.19 | 14.61 | 12.62 | 13.21 | 11.68 | **11.56** |
| Mixup | 90.22 | 85.00 | 57.66 | 25.62 | 21.90 | 16.24 | 12.97 | **10.98** | 11.75 | 14.51 | 13.70 |

We further investigate the effect of $\lambda$ on CIFAR-100 in Table 17. The overall trend closely mirrors our observations on CIFAR-10: PIL remains effective across a broad interval of $\lambda \in [0.3, 0.9]$, indicating that the method is not sensitive to this hyperparameter. Similar to the CIFAR-10 ablations, When $\lambda = 1$ (removing Eq. 3), PIL still provides strong protection, although the performance is slightly less stable across augmentations compared to the best mixed settings ($\lambda = 0.9$). Eq. 3 plays an auxiliary but beneficial role in stabilizing the protection effect.

In particular, $\lambda = 0.9$ yields the most reliable performance across different augmentations. As shown in Table 17, for every augmentation setting, the test accuracy remains below 2.5%, further supporting our choice of $\lambda = 0.9$ as it achieves consistently strong and stable protection on CIFAR-100 as well.

Table 17: CIFAR-100 test accuracy (%) of PIL with varying balancing factor $\lambda$ under different augmentation strategies. Closer to 1% is better.

| Augs | 0.0 | 0.1 | 0.2 | 0.3 | 0.4 | 0.5 | 0.6 | 0.7 | 0.8 | 0.9 | 1.0 |
|------|------|------|------|------|------|------|------|------|------|------|------|
| None | 56.37 | 31.90 | 6.39 | 4.13 | **1.73** | 3.02 | 2.89 | 3.95 | 3.51 | 2.43 | 2.89 |
| Basic | 69.66 | 67.45 | 33.13 | 5.46 | 2.73 | 2.47 | 2.94 | 2.61 | **1.06** | 1.66 | 1.69 |
| Rotation | 67.19 | 61.23 | 11.51 | 4.41 | 2.39 | 2.33 | 1.54 | 1.56 | 1.18 | **1.06** | 1.17 |
| Cutout | 69.89 | 61.97 | 16.95 | 4.21 | 2.31 | 2.22 | 1.69 | 1.24 | **1.13** | 1.94 | 1.59 |
| CutMix | 73.25 | 61.15 | 11.34 | 3.61 | 2.78 | 1.97 | 2.54 | 2.28 | 1.27 | **1.11** | 2.25 |
| Mixup | 69.53 | 60.74 | 17.74 | 6.35 | 5.01 | 3.64 | 2.11 | 1.54 | 1.69 | 1.61 | **1.02** |

## A.13 VISUALIZATION

Figures 7–16 visualize the perturbations and corresponding perturbed datasets produced by PIL, EM, REM, TAP, SP, NTGA, SEP, AR, CUDA, and CUDA* respectively.

A brief comparison of perceptual differences is provided in Table 18. Except for CUDA, which intentionally uses a much larger perturbation budget and consequently produces more noticeable deviations, all other methods exhibit strong imperceptibility. In general, perturbations with PSNR above 30 dB and SSIM above 0.95 are widely regarded as indicators of high-quality images, and nearly all methods fall well within this range.

Table 18: PSNR and SSIM of different unlearnable example methods.

| Metric | EM | REM | TAP | SP | NTGA | SEP | AR | CUDA* | CUDA | PIL (ours) |
|--------|------|------|------|------|------|------|------|------|------|------|
| PSNR | 31.9 | 32.5 | 31.7 | 32.6 | 30.2 | 32.0 | 35.0 | 31.3 | 18.8 | 30.2 |
| SSIM | 0.97 | 0.95 | 0.95 | 0.98 | 0.98 | 0.95 | 0.98 | 0.98 | 0.80 | 0.95 |

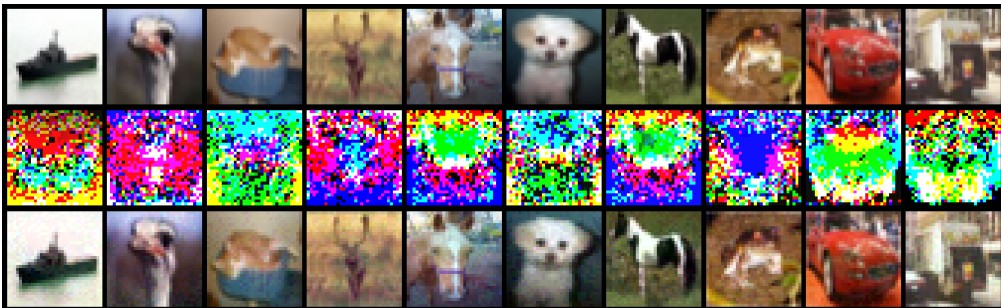

Figure 7: **Samples from PIL constructed CIFAR-10 unlearnable datasets.** We visualize clean images, normalized perturbations and perturbed images (top to bottom).

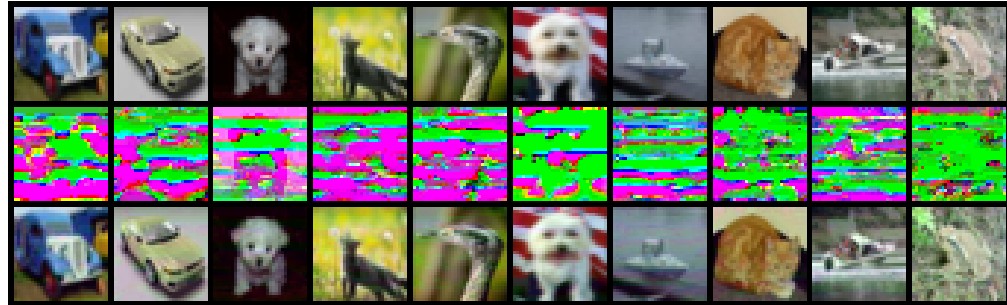

Figure 8: **Samples from EM constructed CIFAR-10 unlearnable datasets.** We visualize clean images, normalized perturbations and perturbed images (top to bottom).

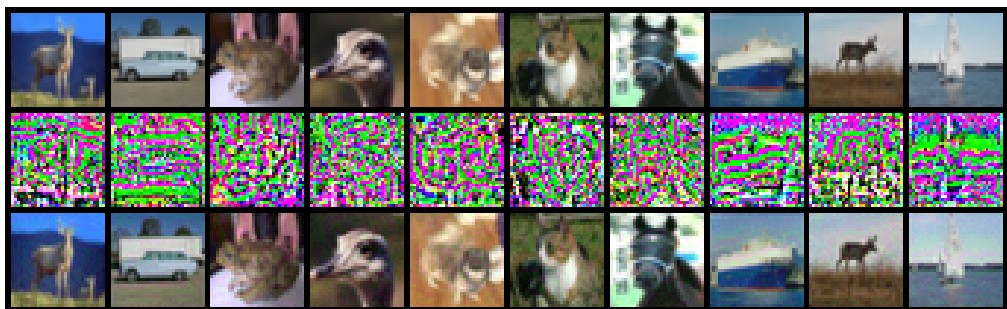

Figure 9: **Samples from REM constructed CIFAR-10 unlearnable datasets.** We visualize clean images, normalized perturbations and perturbed images (top to bottom).

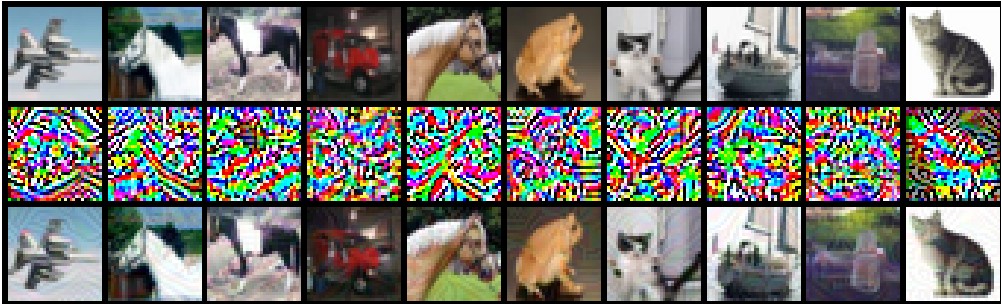

Figure 10: **Samples from TAP constructed CIFAR-10 unlearnable datasets.** We visualize clean images, normalized perturbations and perturbed images (top to bottom).

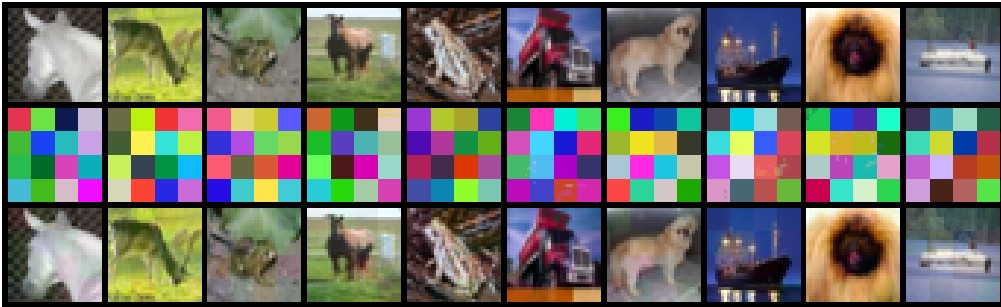

Figure 11: **Samples from SP constructed CIFAR-10 unlearnable datasets.** We visualize clean images, normalized perturbations and perturbed images (top to bottom).

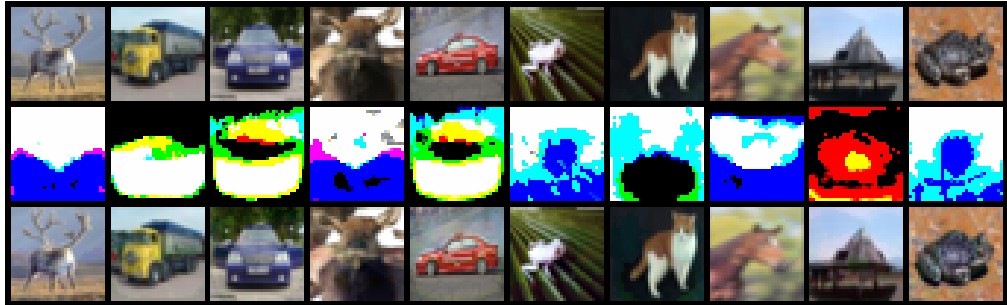

Figure 12: **Samples from NTGA constructed CIFAR-10 unlearnable datasets.** We visualize clean images, normalized perturbations and perturbed images (top to bottom).

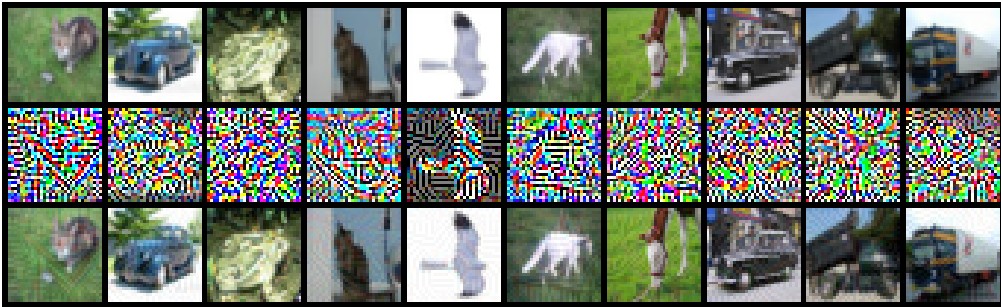

Figure 13: **Samples from SEP constructed CIFAR-10 unlearnable datasets.** We visualize clean images, normalized perturbations and perturbed images (top to bottom).

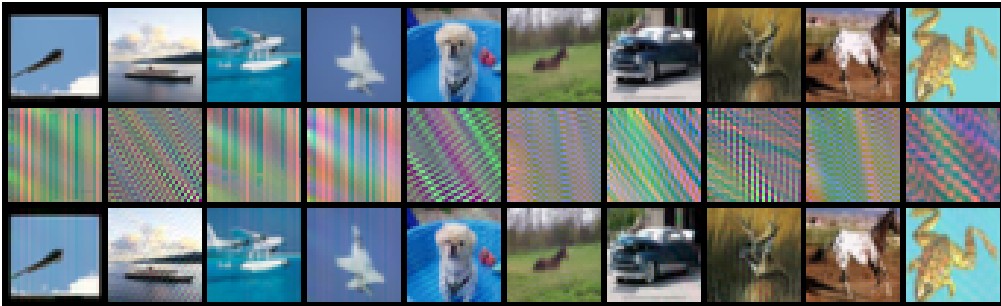

Figure 14: **Samples from AR constructed CIFAR-10 unlearnable datasets.** We visualize clean images, normalized perturbations and perturbed images (top to bottom).

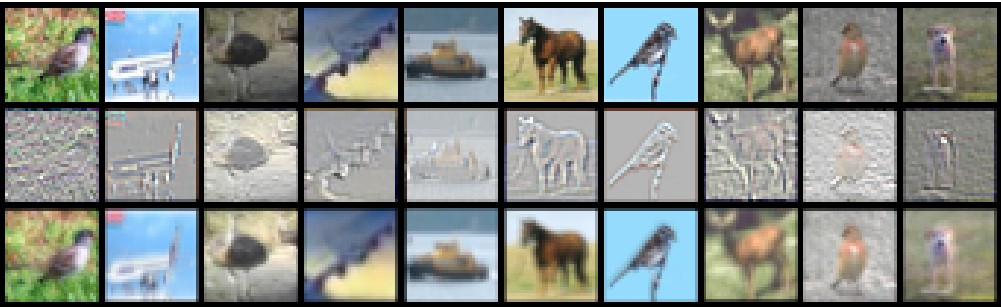

Figure 15: **Samples from CUDA constructed CIFAR-10 unlearnable datasets.** We visualize clean images, normalized perturbations and perturbed images (top to bottom).

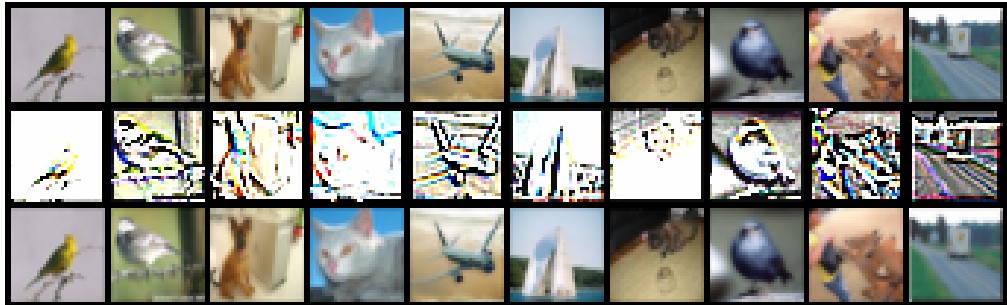

Figure 16: **Samples from CUDA*** ($\epsilon = 8/255$) **constructed CIFAR-10 unlearnable datasets.** We visualize clean images, normalized perturbations and perturbed images (top to bottom).

### A.14  EXPERIMENTAL COMPUTING RESOURCES

All experiments were conducted on the following hardware configuration:

- GPU: NVIDIA RTX A6000 (Memory: 49140MiB)
- CPU: AMD EPYC 7543 32-Core Processor @ 3.7 GHz

### A.15  USE OF LARGE LANGUAGE MODELS

Large Language Models were only used to assist with writing refinement and polishing.

### A.16  MORE DISCUSSION OF SECTION 5

As shown in Tables 19–26, we extend our linearity analysis to several existing unlearnable attack methods, including EM, REM, TAP, SP, NTGA, SEP, and AR. Notably, despite their different designs and objectives, all these methods substantially increase the vulnerability of victim models to FGSM attack, particularly when the perturbation rate reaches 30%. This consistent pattern suggests these methods induce stronger linear behavior in victim models—an effect remarkably similar to our observations in PIL.

This observation is particularly notable because, unlike PIL, these methods do not explicitly aim to make the perturbed datasets linearly separable. Nevertheless, the fact that they all lead to enhanced linearity suggests that this effect may be a fundamental property of unlearnable perturbations. That is, unlearnable attacks may impair learning by implicitly constraining the model to rely on simpler, more linear decision boundaries—thereby limiting its ability to capture meaningful or high-level representations. We leave further exploration of this potential mechanism to future work.

Table 19: Test accuracy and accuracy drop (%) under FGSM attack at different perturbation proportions, ResNet-18 models are trained with mixed clean and PIL perturbed data.

| FGSM Step | 10% | | 20% | | 30% | |
|---|---|---|---|---|---|---|
| | perturbed | clean | perturbed | clean | perturbed | clean |
| 0/255 | 92.33 (-0.0) | 90.83 (-0.0) | 91.7 (-0.0) | 91.47 (-0.0) | 91.97 (-0.0) | 90.87 (-0.0) |
| 1/255 | 61.93 (-30.4) | 61.84 (-28.99) | 59.63 (-32.07) | 61.64 (-29.83) | 49.96 (-42.01) | 60.82 (-30.05) |
| 2/255 | 38.13 (-54.2) | 40.23 (-50.6) | 36.35 (-55.35) | 37.85 (-53.62) | 26.03 (-65.94) | 37.71 (-53.16) |
| 4/255 | 17.39 (-74.94) | 23.79 (-67.04) | 17.68 (-74.02) | 18.45 (-73.02) | 11.75 (-80.22) | 18.75 (-72.12) |
| 8/255 | 7.95 (-84.38) | 14.51 (-76.32) | 9.54 (-82.16) | 9.98 (-81.49) | 9.61 (-82.36) | 11.32 (-79.55) |

| FGSM Step | 40% | | 50% | | 60% | |
|---|---|---|---|---|---|---|
| | perturbed | clean | perturbed | clean | perturbed | clean |
| 0/255 | 91.25 (-0.0) | 90.38 (-0.0) | 89.73 (-0.0) | 87.69 (-0.0) | 88.43 (-0.0) | 84.21 (-0.0) |
| 1/255 | 49.41 (-41.84) | 59.31 (-31.07) | 45.33 (-44.4) | 59.69 (-28.0) | 44.8 (-43.63) | 58.75 (-25.46) |
| 2/255 | 24.56 (-66.69) | 35.91 (-54.47) | 22.13 (-67.6) | 38.23 (-49.46) | 22.6 (-65.83) | 37.24 (-46.97) |
| 4/255 | 9.59 (-81.66) | 16.85 (-73.53) | 9.11 (-80.62) | 21.29 (-66.4) | 8.47 (-79.96) | 19.03 (-65.18) |
| 8/255 | 5.47 (-85.78) | 9.25 (-81.13) | 5.84 (-83.89) | 13.6 (-74.09) | 5.26 (-83.17) | 10.6 (-73.61) |

| FGSM Step | 70% | | 80% | | 90% | |
|---|---|---|---|---|---|---|
| | perturbed | clean | perturbed | clean | perturbed | clean |
| 0/255 | 87.18 (-0.0) | 84.45 (-0.0) | 83.1 (-0.0) | 80.04 (-0.0) | 74.81 (-0.0) | 68.78 (-0.0) |
| 1/255 | 33.38 (-53.8) | 60.7 (-23.75) | 31.02 (-52.08) | 54.82 (-25.22) | 19.6 (-55.21) | 48.56 (-20.22) |
| 2/255 | 15.29 (-71.89) | 39.63 (-44.82) | 13.49 (-69.61) | 35.1 (-44.94) | 7.24 (-67.57) | 32.64 (-36.14) |
| 4/255 | 8.52 (-78.66) | 19.08 (-65.37) | 5.37 (-77.73) | 16.96 (-63.08) | 3.05 (-71.76) | 15.16 (-53.62) |
| 8/255 | 7.48 (-79.7) | 8.42 (-76.03) | 5.77 (-77.33) | 7.66 (-72.38) | 2.61 (-72.2) | 6.34 (-62.44) |

Table 20: Test accuracy and accuracy drop (%) under FGSM attack at different perturbation proportions, ResNet-18 models are trained with mixed clean and EM perturbed data.

| FGSM Step | 10% | | 20% | | 30% | |
|---|---|---|---|---|---|---|
| | perturbed | clean | perturbed | clean | perturbed | clean |
| 0/255 | 92.7 (-0.0) | 91.98 (-0.0) | 91.95 (-0.0) | 90.75 (-0.0) | 88.72 (-0.0) | 89.84 (-0.0) |
| 1/255 | 60.68 (-32.02) | 64.03 (-27.95) | 60.21 (-31.74) | 62.15 (-28.6) | 57.04 (-31.68) | 63.28 (-26.56) |
| 2/255 | 37.33 (-55.37) | 40.85 (-51.13) | 36.32 (-55.63) | 39.01 (-51.74) | 34.71 (-54.01) | 42.14 (-47.7) |
| 4/255 | 19.16 (-73.54) | 19.57 (-72.41) | 17.97 (-73.98) | 19.7 (-71.05) | 17.09 (-71.63) | 22.7 (-67.14) |
| 8/255 | 10.88 (-81.82) | 9.59 (-82.39) | 9.56 (-82.39) | 11.32 (-79.43) | 9.43 (-79.29) | 13.08 (-76.76) |

| FGSM Step | 40% | | 50% | | 60% | |
|---|---|---|---|---|---|---|
| | perturbed | clean | perturbed | clean | perturbed | clean |
| 0/255 | 89.41 (-0.0) | 88.74 (-0.0) | 89.45 (-0.0) | 88.59 (-0.0) | 87.07 (-0.0) | 83.95 (-0.0) |
| 1/255 | 49.56 (-39.85) | 60.28 (-28.46) | 49.18 (-40.27) | 61.69 (-26.9) | 40.95 (-46.12) | 59.62 (-24.33) |
| 2/255 | 28.13 (-61.28) | 38.38 (-50.36) | 26.99 (-62.46) | 40.08 (-48.51) | 23.19 (-63.88) | 40.61 (-43.34) |
| 4/255 | 14.11 (-75.3) | 21.65 (-67.09) | 13.55 (-75.9) | 21.77 (-66.82) | 13.45 (-73.62) | 20.88 (-63.07) |
| 8/255 | 10.06 (-79.35) | 13.3 (-75.44) | 9.15 (-80.3) | 12.94 (-75.65) | 10.74 (-76.33) | 11.15 (-72.8) |

| FGSM Step | 70% | | 80% | | 90% | |
|---|---|---|---|---|---|---|
| | perturbed | clean | perturbed | clean | perturbed | clean |
| 0/255 | 82.99 (-0.0) | 85.25 (-0.0) | 82.15 (-0.0) | 79.22 (-0.0) | 70.37 (-0.0) | 69.36 (-0.0) |
| 1/255 | 37.41 (-45.58) | 59.09 (-26.16) | 30.79 (-51.36) | 55.67 (-23.55) | 17.46 (-52.91) | 48.49 (-20.87) |
| 2/255 | 21.76 (-61.23) | 37.79 (-47.46) | 17.51 (-64.64) | 36.19 (-43.03) | 10.98 (-59.39) | 32.08 (-37.28) |
| 4/255 | 14.01 (-68.98) | 19.07 (-66.18) | 12.10 (-70.05) | 17.08 (-62.14) | 10.47 (-59.90) | 15.54 (-53.82) |
| 8/255 | 10.60 (-72.39) | 9.97 (-75.28) | 9.99 (-72.16) | 7.90 (-71.32) | 9.99 (-60.38) | 6.76 (-62.60) |

Table 21: Test accuracy and accuracy drop (%) under FGSM attack at different perturbation proportions, ResNet-18 models are trained with mixed clean and REM perturbed data.

| FGSM Step | 10% | | 20% | | 30% | |
|---|---|---|---|---|---|---|
| | perturbed | clean | perturbed | clean | perturbed | clean |
| 0/255 | 91.89 (-0.0) | 91.48 (-0.0) | 91.99 (-0.0) | 90.39 (-0.0) | 91.76 (-0.0) | 90.31 (-0.0) |
| 1/255 | 61.25 (-30.64) | 62.4 (-29.08) | 51.75 (-40.24) | 64.96 (-25.43) | 48.75 (-43.01) | 61.69 (-28.62) |
| 2/255 | 37.64 (-54.25) | 40.27 (-51.21) | 27.37 (-64.62) | 41.93 (-48.46) | 24.47 (-67.29) | 38.75 (-51.56) |
| 4/255 | 17.27 (-74.62) | 22.34 (-69.14) | 10.79 (-81.2) | 21.24 (-69.15) | 8.15 (-83.61) | 19.28 (-71.03) |
| 8/255 | 8.63 (-83.26) | 12.66 (-78.82) | 4.48 (-87.51) | 11.01 (-79.38) | 2.65 (-89.11) | 10.59 (-79.72) |

| FGSM Step | 40% | | 50% | | 60% | |
|---|---|---|---|---|---|---|
| | perturbed | clean | perturbed | clean | perturbed | clean |
| 0/255 | 89.14 (-0.0) | 89.9 (-0.0) | 89.03 (-0.0) | 81.87 (-0.0) | 83.13 (-0.0) | 85.78 (-0.0) |
| 1/255 | 53.77 (-35.37) | 62.56 (-27.34) | 47.01 (-42.02) | 57.33 (-24.54) | 40.94 (-42.19) | 63.13 (-22.65) |
| 2/255 | 29.18 (-59.96) | 39.64 (-50.26) | 24.33 (-64.7) | 38.32 (-43.55) | 19.68 (-63.45) | 42.69 (-43.09) |
| 4/255 | 9.15 (-79.99) | 19.37 (-70.53) | 6.98 (-82.05) | 21.74 (-60.13) | 6.08 (-77.05) | 21.83 (-63.95) |
| 8/255 | 3.04 (-86.1) | 9.73 (-80.17) | 1.58 (-87.45) | 13.18 (-68.69) | 3.15 (-79.98) | 10.67 (-75.11) |

| FGSM Step | 70% | | 80% | | 90% | |
|---|---|---|---|---|---|---|
| | perturbed | clean | perturbed | clean | perturbed | clean |
| 0/255 | 86.93 (-0.0) | 85.21 (-0.0) | 79.80 (-0.0) | 81.30 (-0.0) | 74.86 (-0.0) | 70.05 (-0.0) |
| 1/255 | 40.54 (-46.39) | 61.73 (-23.48) | 33.97 (-45.83) | 57.23 (-24.07) | 24.05 (-50.81) | 49.57 (-20.48) |
| 2/255 | 18.96 (-67.97) | 40.93 (-44.28) | 14.79 (-65.01) | 36.64 (-44.66) | 8.25 (-66.61) | 32.49 (-37.56) |
| 4/255 | 4.04 (-82.89) | 20.05 (-65.16) | 3.26 (-76.54) | 17.02 (-64.28) | 1.43 (-73.43) | 13.99 (-56.06) |
| 8/255 | 0.92 (-86.01) | 8.75 (-76.46) | 1.10 (-78.70) | 7.52 (-73.78) | 0.64 (-74.22) | 4.89 (-65.16) |

Table 22: Test accuracy and accuracy drop (%) under FGSM attack at different perturbation proportions, ResNet-18 models are trained with mixed clean and TAP perturbed data.

| FGSM Step | 10% | | 20% | | 30% | |
|---|---|---|---|---|---|---|
| | perturbed | clean | perturbed | clean | perturbed | clean |
| 0/255 | 91.55 (-0.0) | 91.98 (-0.0) | 91.41 (-0.0) | 91.25 (-0.0) | 88.81 (-0.0) | 90.68 (-0.0) |
| 1/255 | 63.31 (-28.24) | 64.64 (-27.34) | 57.59 (-33.82) | 63.54 (-27.71) | 60.78 (-28.03) | 62.58 (-28.1) |
| 2/255 | 41.61 (-49.94) | 40.99 (-50.99) | 33.43 (-57.98) | 39.92 (-51.33) | 38.49 (-50.32) | 39.63 (-51.05) |
| 4/255 | 24.07 (-67.48) | 20.31 (-71.67) | 17.07 (-74.34) | 19.9 (-71.35) | 20.81 (-68.0) | 20.45 (-70.23) |
| 8/255 | 16.37 (-75.18) | 9.87 (-82.11) | 11.98 (-79.43) | 10.5 (-80.75) | 12.04 (-76.77) | 10.7 (-79.98) |

| FGSM Step | 40% | | 50% | | 60% | |
|---|---|---|---|---|---|---|
| | perturbed | clean | perturbed | clean | perturbed | clean |
| 0/255 | 91.03 (-0.0) | 88.92 (-0.0) | 89.79 (-0.0) | 88.12 (-0.0) | 88.19 (-0.0) | 87.69 (-0.0) |
| 1/255 | 46.64 (-44.39) | 64.77 (-24.15) | 41.16 (-48.63) | 64.93 (-23.19) | 42.28 (-45.91) | 59.22 (-28.47) |
| 2/255 | 26.2 (-64.83) | 42.19 (-46.73) | 20.49 (-69.3) | 43.33 (-44.79) | 23.12 (-65.07) | 37.0 (-50.69) |
| 4/255 | 16.5 (-74.53) | 20.59 (-68.33) | 10.66 (-79.13) | 22.23 (-65.89) | 14.65 (-73.54) | 18.41 (-69.28) |
| 8/255 | 14.1 (-76.93) | 9.37 (-79.55) | 10.73 (-79.06) | 10.44 (-77.68) | 14.7 (-73.49) | 10.42 (-77.27) |

| FGSM Step | 70% | | 80% | | 90% | |
|---|---|---|---|---|---|---|
| | perturbed | clean | perturbed | clean | perturbed | clean |
| 0/255 | 87.35 (-0.0) | 81.77 (-0.0) | 83.92 (-0.0) | 80.49 (-0.0) | 80.4 (-0.0) | 71.08 (-0.0) |
| 1/255 | 40.92 (-46.43) | 56.38 (-25.39) | 32.7 (-51.22) | 57.45 (-23.04) | 27.93 (-52.47) | 50.61 (-20.47) |
| 2/255 | 21.43 (-65.92) | 36.29 (-45.48) | 16.29 (-67.63) | 37.07 (-43.42) | 12.39 (-68.01) | 33.84 (-37.24) |
| 4/255 | 10.2 (-77.15) | 18.25 (-63.52) | 11.39 (-72.53) | 17.63 (-62.86) | 6.5 (-73.9) | 14.9 (-56.18) |
| 8/255 | 10.82 (-76.53) | 9.48 (-72.29) | 13.86 (-70.06) | 7.01 (-73.48) | 8.38 (-72.02) | 5.41 (-65.67) |

Table 23: Test accuracy and accuracy drop (%) under FGSM attack at different perturbation proportions, ResNet-18 models are trained with mixed clean and SP perturbed data.

| FGSM Step | 10% | | 20% | | 30% | |
|---|---|---|---|---|---|---|
| | perturbed | clean | perturbed | clean | perturbed | clean |
| 0/255 | 92.66 (-0.0) | 91.16 (-0.0) | 91.88 (-0.0) | 91.78 (-0.0) | 92.33 (-0.0) | 89.82 (-0.0) |
| 1/255 | 58.9 (-33.76) | 63.42 (-27.74) | 63.76 (-28.12) | 60.59 (-31.19) | 54.87 (-37.46) | 64.16 (-25.66) |
| 2/255 | 35.84 (-56.82) | 41.17 (-49.99) | 39.9 (-51.98) | 36.8 (-54.98) | 30.98 (-61.35) | 42.48 (-47.34) |
| 4/255 | 18.53 (-74.13) | 21.44 (-69.72) | 18.47 (-73.41) | 18.17 (-73.61) | 13.09 (-79.24) | 21.27 (-68.55) |
| 8/255 | 11.91 (-80.75) | 12.18 (-78.98) | 7.58 (-84.3) | 9.69 (-82.09) | 5.87 (-86.46) | 10.37 (-79.45) |

| FGSM Step | 40% | | 50% | | 60% | |
|---|---|---|---|---|---|---|
| | perturbed | clean | perturbed | clean | perturbed | clean |
| 0/255 | 89.43 (-0.0) | 89.43 (-0.0) | 88.74 (-0.0) | 87.99 (-0.0) | 85.88 (-0.0) | 84.50 (-0.0) |
| 1/255 | 56.00 (-33.43) | 62.35 (-27.08) | 50.91 (-37.83) | 64.42 (-23.57) | 48.31 (-37.57) | 60.41 (-24.09) |
| 2/255 | 30.29 (-59.14) | 39.99 (-49.44) | 27.20 (-61.54) | 42.72 (-45.27) | 24.29 (-61.59) | 39.60 (-44.90) |
| 4/255 | 10.16 (-79.27) | 21.21 (-68.22) | 10.28 (-78.46) | 21.73 (-66.26) | 8.43 (-77.45) | 20.24 (-64.26) |
| 8/255 | 3.90 (-85.53) | 11.18 (-78.25) | 3.97 (-84.77) | 10.89 (-77.10) | 3.59 (-82.29) | 9.88 (-74.62) |

| FGSM Step | 70% | | 80% | | 90% | |
|---|---|---|---|---|---|---|
| | perturbed | clean | perturbed | clean | perturbed | clean |
| 0/255 | 85.80 (-0.0) | 82.36 (-0.0) | 81.62 (-0.0) | 78.96 (-0.0) | 74.30 (-0.0) | 71.12 (-0.0) |
| 1/255 | 42.79 (-43.01) | 58.12 (-24.24) | 34.20 (-47.42) | 55.18 (-23.78) | 25.37 (-48.93) | 51.03 (-20.09) |
| 2/255 | 20.00 (-65.80) | 37.61 (-44.75) | 14.87 (-66.75) | 35.31 (-43.65) | 7.88 (-66.42) | 34.42 (-36.70) |
| 4/255 | 6.01 (-79.79) | 19.37 (-62.99) | 5.50 (-76.12) | 16.74 (-62.22) | 2.48 (-71.82) | 15.52 (-55.60) |
| 8/255 | 2.57 (-83.23) | 10.08 (-72.28) | 3.37 (-78.25) | 8.00 (-70.96) | 2.18 (-72.12) | 5.30 (-65.82) |

Table 24: Test accuracy and accuracy drop (%) under FGSM attack at different perturbation proportions, ResNet-18 models are trained with mixed clean and NTGA perturbed data.

| FGSM Step | 10% | | 20% | | 30% | |
|---|---|---|---|---|---|---|
| | perturbed | clean | perturbed | clean | perturbed | clean |
| 0/255 | 91.48 (-0.0) | 91.96 (-0.0) | 91.83 (-0.0) | 90.71 (-0.0) | 91.44 (-0.0) | 89.61 (-0.0) |
| 1/255 | 60.97 (-30.5) | 60.71 (-31.2) | 59.97 (-31.9) | 62.45 (-28.3) | 55.22 (-36.2) | 60.93 (-28.7) |
| 2/255 | 38.66 (-52.8) | 35.99 (-56.0) | 36.44 (-55.4) | 40.56 (-50.1) | 31.75 (-59.7) | 39.20 (-50.4) |
| 4/255 | 20.22 (-71.3) | 17.40 (-74.6) | 17.98 (-73.8) | 21.59 (-69.1) | 14.82 (-76.6) | 23.08 (-66.5) |
| 8/255 | 12.23 (-79.2) | 8.92 (-83.0) | 10.42 (-81.4) | 11.94 (-78.8) | 8.15 (-83.3) | 14.50 (-75.1) |

| FGSM Step | 40% | | 50% | | 60% | |
|---|---|---|---|---|---|---|
| | perturbed | clean | perturbed | clean | perturbed | clean |
| 0/255 | 89.88 (-0.0) | 91.05 (-0.0) | 89.61 (-0.0) | 85.92 (-0.0) | 89.10 (-0.0) | 86.60 (-0.0) |
| 1/255 | 52.60 (-37.3) | 53.55 (-37.5) | 48.98 (-40.6) | 62.80 (-23.1) | 41.87 (-47.2) | 59.27 (-27.3) |
| 2/255 | 27.72 (-62.2) | 31.92 (-59.1) | 24.94 (-64.7) | 42.29 (-43.6) | 18.35 (-70.8) | 37.53 (-49.1) |
| 4/255 | 9.71 (-80.2) | 17.07 (-74.0) | 9.96 (-79.6) | 22.28 (-63.6) | 6.29 (-82.8) | 19.58 (-67.0) |
| 8/255 | 4.12 (-85.8) | 10.27 (-80.8) | 4.90 (-84.7) | 11.75 (-74.2) | 2.93 (-86.2) | 10.82 (-75.8) |

| FGSM Step | 70% | | 80% | | 90% | |
|---|---|---|---|---|---|---|
| | perturbed | clean | perturbed | clean | perturbed | clean |
| 0/255 | 85.00 (-0.0) | 80.00 (-0.0) | 81.13 (-0.0) | 80.53 (-0.0) | 71.49 (-0.0) | 70.04 (-0.0) |
| 1/255 | 36.41 (-48.6) | 54.38 (-25.6) | 28.89 (-52.2) | 55.84 (-24.7) | 16.54 (-54.9) | 50.12 (-19.9) |
| 2/255 | 17.00 (-68.0) | 35.16 (-44.8) | 11.80 (-69.3) | 36.58 (-43.9) | 5.07 (-66.4) | 34.15 (-35.9) |
| 4/255 | 6.97 (-78.0) | 19.08 (-60.9) | 4.26 (-76.9) | 18.92 (-61.6) | 1.89 (-69.6) | 16.47 (-53.6) |
| 8/255 | 3.74 (-81.3) | 11.41 (-68.6) | 2.36 (-78.8) | 10.94 (-69.6) | 1.54 (-69.9) | 6.99 (-63.1) |

Table 25: Test accuracy and accuracy drop (%) under FGSM attack at different perturbation proportions, ResNet-18 models are trained with mixed clean and SEP perturbed data.

| FGSM Step | 10% | | 20% | | 30% | |
|---|---|---|---|---|---|---|
| | perturbed | clean | perturbed | clean | perturbed | clean |
| 0/255 | 91.53 (-0.00) | 92.1 (-0.00) | 91.18 (-0.00) | 90.44 (-0.00) | 91.05 (-0.00) | 88.65 (-0.00) |
| 1/255 | 69.79 (-21.74) | 62.38 (-29.72) | 68.48 (-22.70) | 63.25 (-27.19) | 61.32 (-29.73) | 65.0 (-23.65) |
| 2/255 | 48.68 (-42.85) | 37.58 (-54.52) | 46.06 (-45.12) | 41.26 (-49.18) | 39.06 (-51.99) | 43.53 (-45.12) |
| 4/255 | 26.98 (-64.55) | 17.34 (-74.76) | 25.02 (-66.16) | 21.16 (-69.28) | 21.44 (-69.61) | 22.4 (-66.25) |
| 8/255 | 14.7 (-76.83) | 9.36 (-82.74) | 13.11 (-78.07) | 11.37 (-79.07) | 13.2 (-77.85) | 9.17 (-79.48) |

| FGSM Step | 40% | | 50% | | 60% | |
|---|---|---|---|---|---|---|
| | perturbed | clean | perturbed | clean | perturbed | clean |
| 0/255 | 90.99 (-0.00) | 90.5 (-0.00) | 90.01 (-0.00) | 88.92 (-0.00) | 89.25 (-0.00) | 87.35 (-0.00) |
| 1/255 | 55.79 (-35.20) | 58.91 (-31.59) | 51.25 (-38.76) | 59.76 (-29.16) | 44.16 (-45.09) | 60.23 (-27.12) |
| 2/255 | 31.77 (-59.22) | 35.18 (-55.32) | 27.8 (-62.21) | 38.19 (-50.73) | 23.96 (-65.29) | 38.53 (-48.82) |
| 4/255 | 14.04 (-76.95) | 17.39 (-73.11) | 12.48 (-77.53) | 19.33 (-69.59) | 12.5 (-76.75) | 20.32 (-67.03) |
| 8/255 | 9.34 (-81.65) | 9.37 (-81.13) | 11.15 (-78.86) | 11.26 (-77.66) | 12.44 (-76.81) | 11.82 (-75.53) |

| FGSM Step | 70% | | 80% | | 90% | |
|---|---|---|---|---|---|---|
| | perturbed | clean | perturbed | clean | perturbed | clean |
| 0/255 | 86.99 (-0.00) | 86.04 (-0.00) | 85.7 (-0.00) | 79.49 (-0.00) | 82.47 (-0.00) | 77.96 (-0.00) |
| 1/255 | 46.02 (-40.97) | 61.69 (-24.35) | 38.75 (-46.95) | 55.84 (-23.65) | 31.88 (-50.59) | 53.07 (-24.89) |
| 2/255 | 24.37 (-62.62) | 39.41 (-46.63) | 19.4 (-66.30) | 35.67 (-43.82) | 13.68 (-68.79) | 31.79 (-46.17) |
| 4/255 | 12.08 (-74.91) | 18.8 (-67.24) | 9.02 (-76.68) | 16.13 (-63.36) | 6.25 (-76.22) | 13.3 (-64.66) |
| 8/255 | 12.66 (-74.33) | 7.41 (-78.63) | 12.23 (-73.47) | 7.08 (-72.41) | 7.95 (-74.52) | 4.74 (-73.22) |

Table 26: Test accuracy and accuracy drop (%) under FGSM attack at different perturbation proportions, ResNet-18 models are trained with mixed clean and AR perturbed data.

| FGSM Step | 10% | | 20% | | 30% | |
|---|---|---|---|---|---|---|
| | perturbed | clean | perturbed | clean | perturbed | clean |
| 0/255 | 92.96 (-0.00) | 92.1 (-0.00) | 91.91 (-0.00) | 90.44 (-0.00) | 92.37 (-0.00) | 88.65 (-0.00) |
| 1/255 | 57.59 (-35.37) | 62.38 (-29.72) | 65.49 (-26.42) | 63.25 (-27.19) | 48.18 (-44.19) | 65.0 (-23.65) |
| 2/255 | 33.49 (-59.47) | 37.58 (-54.52) | 42.05 (-49.86) | 41.26 (-49.18) | 26.09 (-66.28) | 43.53 (-45.12) |
| 4/255 | 16.91 (-76.05) | 17.34 (-74.76) | 20.4 (-71.51) | 21.16 (-69.28) | 14.84 (-77.53) | 22.4 (-66.25) |
| 8/255 | 10.9 (-82.06) | 9.36 (-82.74) | 10.26 (-81.65) | 11.37 (-79.07) | 10.28 (-82.09) | 9.17 (-79.48) |

| FGSM Step | 40% | | 50% | | 60% | |
|---|---|---|---|---|---|---|
| | perturbed | clean | perturbed | clean | perturbed | clean |
| 0/255 | 91.13 (-0.00) | 90.5 (-0.00) | 90.62 (-0.00) | 88.92 (-0.00) | 87.8 (-0.00) | 87.35 (-0.00) |
| 1/255 | 50.44 (-40.69) | 58.91 (-31.59) | 36.31 (-54.31) | 59.76 (-29.16) | 41.61 (-46.19) | 60.23 (-27.12) |
| 2/255 | 35.2 (-55.93) | 35.18 (-55.32) | 21.5 (-69.12) | 38.19 (-50.73) | 26.92 (-60.88) | 38.53 (-48.82) |
| 4/255 | 19.69 (-71.44) | 17.39 (-73.11) | 13.91 (-76.71) | 19.33 (-69.59) | 12.92 (-74.88) | 20.32 (-67.03) |
| 8/255 | 10.5 (-80.63) | 9.37 (-81.13) | 9.92 (-80.70) | 11.26 (-77.66) | 11.23 (-76.57) | 11.82 (-75.53) |

| FGSM Step | 70% | | 80% | | 90% | |
|---|---|---|---|---|---|---|
| | perturbed | clean | perturbed | clean | perturbed | clean |
| 0/255 | 87.39 (-0.00) | 86.04 (-0.00) | 84.96 (-0.00) | 79.49 (-0.00) | 79.55 (-0.00) | 77.96 (-0.00) |
| 1/255 | 40.39 (-47.00) | 61.69 (-24.35) | 28.43 (-56.53) | 55.84 (-23.65) | 15.39 (-64.16) | 53.07 (-24.89) |
| 2/255 | 22.92 (-64.47) | 39.41 (-46.63) | 19.35 (-65.61) | 35.67 (-43.82) | 10.67 (-68.88) | 31.79 (-46.17) |
| 4/255 | 17.61 (-69.78) | 18.8 (-67.24) | 13.27 (-71.69) | 16.13 (-63.36) | 10.04 (-69.51) | 13.3 (-64.66) |
| 8/255 | 9.97 (-77.42) | 7.41 (-78.63) | 11.05 (-73.91) | 7.08 (-72.41) | 9.32 (-70.23) | 4.74 (-73.22) |