# OpenReview forum: "Perturbation-Induced Linearization: Constructing Unlearnable Data with Solely Linear Classifiers"
_ICLR.cc/2026/Conference — ICLR 2026 Poster_

### Official Review · Reviewer_QLdv · 2025-10-28

**Soundness:** 2
**Presentation:** 2
**Contribution:** 1
**Rating:** 4
**Confidence:** 4

**Summary:**

This paper proposed a new algorithm to create unlearnable data by inducing linearization to models through the crafted perturbations.

**Strengths:**

1. The proposed algorithm works well on making unlearnable datasets for CIFAR-10/100, SVHN, and ImageNet.
2. PIL is the most time-efficient surrogate-based model.

**Weaknesses:**

1. The motivation and mechanism of the proposed algorithm are obfuscating. The combination and decomposition between $\delta$ and $\delta_1+\delta_2$, is problematic and not convincing. Equ (8) and Line 212 are very confusing.
2. Compared to all baseline methods, the performance of the proposed method is not consistently best and shows incremental improvements. Although it's much efficient than other surrogate model-based methods, the contribution is not that significant to me.
3. Theorem 1 is self-contradicted as a large $\alpha$ will lead to abrupt accuracy drop.

**Questions:**

See weakness

---

> ### Author Response · Authors · 2025-11-25
> **Response to Reviewer QLdv**
>
> ## **Response to Reviewer QLdv**
> We sincerely thank the reviewer for the insightful comments and suggestions. Below we address each concern in detail.
>
> ### **Weakness 1**
> **Question:**
> *"The motivation and mechanism of the proposed algorithm are obfuscating."*
>
> **Response:**
> Thank you for raising this point. We clarify the motivation and mechanisms behind PIL as follows:
>
> Our goal is simple: **we want the model to learn a shortcut induced by the perturbation, rather than the true semantic features**. PIL achieves this through two components:
>
> 1. **Eq.(4) — Shortcut Learning:** This term constructs a linear correspondence between the perturbation and the label.
>
> 2. **Eq.(3) — Semantic Obfuscation:** This term suppresses the true informative features.
>
> Together, these two components achieve the intended effect: strengthen shortcut learning while weakening real features.
>
> **Question:**
> *"The combination and decomposition between $\delta$ and $\delta_1+\delta_2$ is problematic and not convincing, and Eq (8) and Line 212 are very confusing."*
>
> **Response:**
>
> We first clarify the meaning of the symbols and the joint optimization:
>
> * **$\delta_1$** corresponds to Eq.(3) (Semantic Obfuscation)
> * **$\delta_2$** corresponds to Eq.(4) (Shortcut Learning)
> * **Joint optimization:** combining $\delta_1$ and $\delta_2$ via their sum $\delta$ is a conventional approach and does not introduce issues.
>
> We believe the confusion stems from why **Eq.(3) and Eq.(7) use $x - \delta$**, while **Eq.(4) uses $+\delta_2$ instead of $-\delta_2$**.
>
> **Regarding the sign choice:**
> Eq.(8) shows that **$\delta_2$ is strongly negatively correlated with the label**. This is still easily learnable— even a linear classifier could fit it simply by flipping the sign of its weight vector. A deep model, with stronger fitting capacity, learns this correspondence even more easily.
>
> **Why do we intentionally choose the negative correlation?**
> Because our perturbation generator is a pretrained linear model, even a simple linear classifier can achieve ~40% accuracy on CIFAR-10. Choosing the negative direction helps prevent the downstream model from reaching similar accuracy.
>
>
> We verified this design choice experimentally. We define **PIL⁺** as a variant where Eq.(3) and Eq.(7) use **$x + \delta$** instead of **$x - \delta$**.
>
> Under CIFAR-10 with Basic augmentation:
>
> Method|Acc
> -|-
> PIL⁺|32.37%
> PIL|12.87%
>
> The original PIL is clearly more robust under augmentation, validating our design.
>
> ---
>
> ### **Weakness 2**
> **Question:**
> *"The proposed method is not consistently best and shows incremental improvements."*
>
> **Response:**
> We acknowledge that PIL does not always strictly outperform all baselines. However, PIL delivers **comparable or better performance with orders-of-magnitude lower computational cost**. Compared to AR and SEP, PIL requires **hundreds to thousands of times less runtime**, while achieving similar effectiveness.
>
> Moreover, efficiency is not our only contribution. As noted by Reviewer VbQk, our paper also reports:
>
> > “A new interesting finding: all unlearnable methods make models more vulnerable to FGSM, which the authors use as a proxy for showing that models trained on unlearnable data become more linear.”
>
> We believe the **more fundamental contribution** of our work lies in uncovering a **key mechanism** behind unlearnable examples:
>
> - They induce deep models to behave more like **linear models**, which may reduce the ability of deep models to learn meaningful representations.
>
> We believe this insight will help deepen understanding of why certain perturbations are effective, and provides a new direction for future research in this area.
>
> ---
>
> ### **Weakness 3**
> **Question:**
> *"Theorem 1 is self-contradicted as a large $\alpha$ will lead to abrupt accuracy drop."*
>
> **Response:**
> The confusion may comes from a labeling issue in Fig.3. We have now renamed the blue curve as **“mixed”** to clarify that:
>
> * **blue curve**: training with a mixture of clean samples and PIL samples
> * **red curve**: training with only clean samples
>
> Example: at $\alpha= 0.8$, the blue point corresponds to **80% PIL + 20% clean data**, and the red point corresponds to **only the 20% clean data**.
>
> Theorem 1 states that **unlearnable examples neither help nor hinder the learning process on clean samples**. This is consistent with Fig.3:
>
> * As $\alpha$ increases, the **amount of clean data decreases**, causing **both** curves to drop sharply. This is a data-scarcity effect, not a failure of Theorem 1.
>
> * Importantly, for the same amount of clean data, the red and blue curves behave similarly, **supporting** Theorem 1.
>
> Thus, Theorem 1 is not self-contradictory; the performance drop at large $\alpha$ is due to insufficient clean data, consistent with the theorem.
>
> In the revised version, we have rewritten and expanded Section 6 to provide a clearer exposition of Theorem 1.

---

### Official Review · Reviewer_VbQk · 2025-10-29

**Soundness:** 3
**Presentation:** 4
**Contribution:** 3
**Rating:** 8
**Confidence:** 5

**Summary:**

The authors propose that unlearnable examples work because they induce model linearity. Using existing unlearnable dataset methods, they show that models trained on unlearnable data are more linear. Model linearity is measured by the attack success rate of FGSM attacks. Using this hypothesis, they design a new loss function that optimizes effective unlearnable data perturbations in less time than other approaches due to the tiny number of surrogate parameters.

**Strengths:**

In my batch so far, this is the most well-written paper. The writing/math is very clear, and some sections are better than prior conference work which introduce unlearnable example methods (especially wrt the defense-attacker definition and motivation of the problem). I can tell the authors were careful in how they designed notation to explain their method. It was a fun read.

There are a number of novel contributions:
1. a simple (2 loss components optimized with SGD) loss function that optimizes effective perturbations (across datasets). See Table 1. The perturbations also work across common augmentations (Table 2).
2. a new interesting finding: all unlearnable methods make models more vulnerable to FGSM, which the authors use as a proxy for showing that models trained on unlearnable data become more linear. This motivates their approach. See Tables 6-7.
3. the authors tie together loose ends on why unlearnable examples work. Linear separability of perturbations [2] was initially thought to be the mechanism, but there were counter-examples like AR unlearnable examples. Their linear behavior hypothesis is backed up by their FGSM experiments, but they also consider alternatives like the possibility that networks simply learn a correspondence between perturbations and labels (Section 4.4 and Table 5). I could see their FGSM approach inspiring future work to check other unlearnable data methods.

The strengths of this work outweigh the weaknesses because I feel it gives a convincing argument on why unlearnable data works, and there continues to be work in this area.

**Weaknesses:**

1. For common defenses (Section 4.2.1), ISS [1] is likely the most important, reasonable, and cheap defense against unlearnable examples. I am particularly interested in an evaluation of *just ISS* (different JPEG compression qualities should be tried: 0.9, 0.8, 0.7, etc) instead of all the other "defenses" (cutout, cutmix, mixup, etc.) because JPEG has been shown to be so effective but the augmentations provided in Table 2 are a good start (and can still be in appendix). The ISS [1] paper broke a number of existing "unlearnable datasets" but this submission only considers augmentations outside of JPEG. Other augmentations can't really be considered a defense after the ISS paper.
2. Additionally, being a linear perturbation, I wonder if the orthogonal projection [3] defense (Section 4.4 of [3]) would work against these perturbations. The authors of [3] argue linear perturbations can be easily broken. It would involve training a linear model on PIL data, then projecting data orthogonal to that learned linear model.
3. One of the main claims is that this work reveals a fundamental property of unlearnable examples: "they cannot substantially reduce test accuracy when only part of the dataset is perturbed". But this has already been shown in [1] Table 2, where with only 20% clean data, training on the sample-wise poison gets 86.85% accuracy. Only at 100% of unlearnable data do we see the more than 70% drop. I'm not sure if that can be considered a contribution of this work.

[1] Unlearnable Examples: Making Personal Data Unexploitable. Huang et al., 2021

[3] What can we learn from unlearnable datasets?, Sandoval-Segura et al., 2023

I'd note that addressing 1. and 2. does not take away from Section 5's findings.

**Questions:**

1. If the unlearnable data (original image + delta) is linearly separable, is the model trained on the unlearnable data expected to be linear too? If so, the results here remind me of [2] (but I understand the claim here is different bc [2] is about separability of perturbations, and the argument here is about the learned function)
2. Was an ablation of the the loss components was considered? I wonder how much less effective the method would be without Eq. 3 (lambda  = 1) or without Eq. 4 (lambda = 0). Based on your analysis, I'd expect Eq. 4 loss to matter more, but it'd be great to get an answer on that. (Doesn't have to be on all datasets or models, just one example to give a sense of what the answer is)
3. Were there other proxies you considered that could test model linearity?
4. Does it make sense to caveat a little the claim of Line 410? It seems like SEP doesn't have such a wide gap with clean/perturbed models on FGSM?

[2] Availability attacks create shortcuts. Yu et al., 2022

---

> ### Author Response · Authors · 2025-11-25
> **Response to Reviewer VbQk (Weakness)**
>
> # **Response to Reviewer VbQk (Weakness)**
> We sincerely thank the reviewer for the insightful comments and suggestions. Below we address each concern in detail.
> ## **Weakness1**
> **Question:** *"I am particularly interested in an evaluation of just ISS (different JPEG compression qualities should be tried: 0.9, 0.8, 0.7, etc)"*
>
> **Response:** We fully agree that JPEG compression is a very important defense against unlearnable examples. Following your suggestion, we added a dedicated experiment in the main text (**Table 3**) that systematically evaluates different unlearnable methods under JPEG compression with various quality factors.
>
> For clarity, we reproduce Table.3 here and highlight the Top-2 unlearnable examples (lowest test accuracy) for each JPEG quality in **bold**:
>
>
> |Method|JPEG90|JPEG80|JPEG70|JPEG60|JPEG50|JPEG40|JPEG30| JPEG20 | JPEG10 |
> |-|-|-|-|-|-|-|-|-|-|
> |Clean|90.99|88.61|89.74|89.43|87.87|88.44|87.67|87.06|83.90|
> |EM|25.75| **34.04** | 44.15  | 51.58  | 55.12  | 61.13  | 70.94  | 72.50  | 80.94  |
> | REM       | 67.21  | 79.92  | 81.64  | 81.22  | 82.85  | 83.69  | 83.56  | 84.58  | 82.79  |
> | TAP       | **20.10** | 42.87  | 64.80  | 72.21  | 78.69  | 82.14  | 84.01  | 84.12  | 83.14  |
> | NTGA      | 41.57  | 52.14  | 53.70  | 55.97  | 63.48  | 62.24  | 67.31  | 72.97  | **74.64** |
> | SEP       | **12.18** | 49.66  | 64.09  | 76.46  | 83.08  | 85.99  | 87.08  | 86.21  | 82.43  |
> | AR        | 53.53  | 69.36  | 79.61  | 84.66  | 86.53  | 87.24  | 87.64  | 86.74  | 83.35  |
> | SP        | 26.17  | **31.74** | **29.94** | **33.10** | **34.72** | **40.50** | **40.20** | **54.04** | 79.34  |
> | PIL (ours)| 35.26  | 36.97  | **41.55** | **43.64** | **50.87** | **52.05** | **58.37** | **67.89** | **76.71** |
>
>
> **SP** and **PIL** (both linear‑based methods) exhibit strong robustness under a wide quality range of JPEG compression.
> **These results may suggest that constructing unlearnable examples based on linear separability is a more fundamental approach.**
>
> We have explicitly added this discussion to **Section 4.2.1**.
>
>
> ---
>
> ## **Weakness2**
>
> **Question:**
> *"Additionally, being a linear perturbation, I wonder if the orthogonal projection defense would work against these perturbations."*
>
>
> **Response:**
> Thank you for this insightful suggestion. Following, we implemented an orthogonal projection defense to our PIL perturbations:
>
> 1. Train a linear classifier on the PIL‑protected dataset (CIFAR‑10).
> 2. For each data point, project it onto the subspace orthogonal to the learned linear classifier.
> 3. Train a standard deep model on this “orthogonally projected” dataset.
>
> We found:
>
> - Without any defense, on PIL‑protected CIFAR‑10, the model accuracy is **14.70%**.
> - With orthogonal projection, accuracy is restored to **40.86%**.
>
> Thus, **orthogonal projection can weaken our protection but does not fully restore clean performance**, which is around 90% on CIFAR‑10 in our setting.
>
> ---
>
> ## **Weakness3**
> **Question:**
> *"One of the main claims is that this work reveals a fundamental property of unlearnable examples: "they cannot substantially reduce test accuracy when only part of the dataset is perturbed". But this has already been shown in [1] Table 2 ... I'm not sure if that can be considered a contribution of this work."*
>
>
> **Response:**
> Thank you for this careful remark and for pointing us back to [1, Table 2].
>
> We fully acknowledge that Huang et al. [1] were the first to observe this phenomenon. As they noted in their paper, “(This phenomenon) may not be a failure of the error‑minimizing noise”. Our intention was not to claim this finding as entirely new, but to build upon their qualitative insight in two specific ways:
>
> First, we conducted more **quantitative experiments** (Appendix 10.2, Fig. 4) to measure how much accuracy gain from perturbed samples corresponds to the gain from a given number of clean samples—making the tradeoff between clean and poisoned data explicit.
>
> More importantly, our key original contribution lies in uncovering a general **gradient orthogonality phenomenon** (Appendix 4.1, Table 10): across all unlearnable example methods we tested, after a few training epochs, the gradients from perturbed samples become nearly orthogonal (cosine similarity close to 0) to those from clean samples. This observation helps explain why unlearnable examples do not substantially interfere with learning from clean data, and suggests that the sharp accuracy drop at high perturbation ratios stems mainly from the scarcity of clean examples—not because unlearnable examples only work when they dominate the dataset.
>
> In the revised version, we have rewritten and expanded Section 6 to more clearly position Huang et al.’s qualitative finding, present our quantitative results, and highlight our novel gradient orthogonality analysis as a new explanation for the behavior across different unlearnable methods.
>
> [1] Unlearnable Examples: Making Personal Data Unexploitable. Huang et al., 2021

---

> ### Author Response · Authors · 2025-11-25
> **Response to Reviewer VbQk (Questions)**
>
> # **Response to Reviewer VbQk (Questions)**
>
> ## **Q1 & Q3： Linearity**
> Before directly answering Q1, we first address Q3 (other linearity proxies).
>
> We use FGSM as a proxy for **local linearity**, since FGSM essentially exploits the model’s first-order approximation, and models that behave more linearly locally tend to align more closely with this first-order prediction.
>
> ### **Answer to Q3:**
>
> *" Other proxies for linearity."*
>
> Beyond local linearity, we have also considered two complementary proxies:
>
> 1. **Global linear fit**
>    We sample random inputs $x$, collect deep model outputs $y$, fit a linear regression $x \mapsto y$, and compute the coefficient of determination:
>    $$
>    R^2_{\text{global}}.
>    $$
>
> 2. **Linearity along random directions**
>    We sample random directions in the input space, take evenly spaced points, regress positional indices against model outputs, and average:
>    $$
>    R^2_{\text{dir}}.
>    $$
>
> The combined results appear below:
>
> Method| $R^2_{\text{global}}$ | $R^2_{\text{dir}}$
> -|-|-
> CLEAN|0.348| 0.373
> EM |0.626| **0.788**
> REM|0.269|**0.652**
> TAP|0.326| **0.435**
> SP |0.690| **0.875**
> NTGA|**0.746**| 0.603
> SEP|**0.530**| 0.283
> AR |0.135| **0.631**
> PIL|**0.598**| 0.580
>
> Across many methods (EM, SP, NTGA, PIL), **both measures exceed the clean baseline**, indicating increased global linearity. For the remaining methods (REM, TAP, SEP, AR), the two proxies do not fully agree, reflecting the fact that global linearity metrics themselves may be inconsistent when applied to complex deep networks.
>
> Nevertheless, whether the increase appears in the overall regression metric $R^2_{\text{global}}$ or in the direction-based metric $R^2_{\text{dir}}$, **any improvement over the clean baseline already indicates a shift toward more linear behavior**, even though the two metrics do not always change in perfect synchrony.
>
> Because these global metrics can be inconsistent, we turn to FGSM-based probing for **local** linearity, which provides more stable and reliable insight into how unlearnable examples promote linearization.
>
>
> ### **Answer to Q1:**
> *"Is a model trained on linearly separable unlearnable data expected to be linear?"*
>
> Empirically, **no**, at least not in a global sense.
>
> For methods like **SP** and **PIL**, the perturbed data are (by design) linearly separable. However, for SP and PIL, the **global** linear fits (either over full input space or along random directions) do not support such a hypothesis.
>
> We conjecture that this mismatch may relate to **Batch Normalization** and non‑linear activations. Even if all pre‑BN activations in some layers are positive, BatchNorm will normalize them, which may produce some negative values and thereby trigger nonlinear ReLU behavior. We acknowledge, however, this may be just one factor.
>
> ---
>
> ## **Q2: Ablation**
> Thank you for pointing this out; we agree this is important. We have added an ablation study in **Appendix 12**, where we compare:
>
> -  $\lambda= 0$ : Eq. (3) only
> -  $\lambda= 1$ : Eq. (4) only
>
> To better illustrate the contribution of each component, we extract key results from our full ablation table (test accuracy, %):
>
> |Augs|$\lambda= 0$|$\lambda= 0.9$|$\lambda= 1$|
> |-|-|-|-|
> |None|82.97|17.08|18.76|
> |Basic|90.71|15.98|13.12|
> |Rotation|88.56|17.41|20.10|
> |Cutout|92.54|12.34|21.58|
> |CutMix|92.20|11.68|11.56|
> |Mixup|90.22|14.51|13.70|
>
> Your intuition is correct: **Eq. (4) (the linearity‑inducing component) matters more**.
>
> With $\lambda= 1$  (Eq. (4) only), PIL achieves strong protective performance, showing that inducing linearity is the main driver of our method.
>
> However, **Eq. (3) is still important**: Incorporating Eq. (3) provides additional stability and robustness to the protection. With $\lambda = 0.9$, our method consistently reduces model accuracy below 20% across all augmentation settings. We now explicitly report and discuss this in **Appendix 12**.
>
> ---
>
> ## **Q4: SEP’s smaller FGSM gap**
> Thank you for highlighting this nuanced point. We agree with your observation: relative to the other methods, SEP (which aggregates multiple surrogate models) shows a smaller increase in FGSM vulnerability. A plausible explanation is that the perturbations produced by multiple surrogates are more complex and may therefore encourage weaker linearization than the other approaches.
>
> For our main claim, the key takeaway from Table 8 is that **every entry is positive**: across all unlearnable methods, training on perturbed data yields higher FGSM success rates than training on clean data. This consistent sign pattern supports the view that increased linearity is a mechanism shared by all unlearnable examples we tested, rather than a property unique to PIL.
>
> In the revision, we have:
> - Softened the wording around the claim near Line 410.
> - Explicitly guided readers to focus on the sign (all entries positive).

---

> ### Author Response · Authors · 2025-11-25
> **A Note on Response Length**
>
> Due to the 5,000-character limit per response, we have split our rebuttal into two separate messages—one addressing weaknesses and another addressing questions. We apologize for any inconvenience this may cause. We sincerely appreciate the reviewer’s thoughtful insights and recognition of our work. For more detailed discussion and clarifications, please refer to the revised PDF. We hope this arrangement is acceptable.

---

### Official Review · Reviewer_fPn9 · 2025-10-29

**Soundness:** 3
**Presentation:** 3
**Contribution:** 3
**Rating:** 6
**Confidence:** 2

**Summary:**

This paper introduces Perturbation-Induced Linearization (PIL), an efficient method that generates perturbations using only linear surrogate models. PIL achieves comparable or better performance than existing surrogate-based methods while reducing computational time (reported ~40s on CIFAR-10). It also uncovers a key mechanism behind unlearnable examples: they induce deep models
to behave more like linear models, which may reduce their capacity to learn meaningful
representations. This paper also analyzes why accuracy is harder to suppress when only a partial fraction of the training set is perturbed.

**Strengths:**

1. Simplicity. PIL only use only linear surrogate models. On CIFAR-10, the reported generation time is under one GPU-minute.

2. Efficiency. PIL remains effective under various data augmentation strategies and adversarial training.

**Weaknesses:**

1. Please include results for larger initial clean ratios (η), e.g., 0.8, to validate how perturbed samples contribute to accuracy improvements.

2. This paper primarily focuses on methods developed up to 2022. It would be helpful to include comparisons with more recent work—such as CUDA [1] and UGEs [2] in effectiveness and generation time.

[1] Vinu Sankar Sadasivan, Mahdi Soltanolkotabi, and Soheil Feizi. Cuda: Convolution-based unlearnable datasets. In Proceedings of the IEEE/CVF Conference on Computer Vision and Pattern Recognition, pages 3862–3871, 2023.

[2] Ye J, Wang X. Ungeneralizable examples[C]//Proceedings of the IEEE/CVF Conference on Computer Vision and Pattern Recognition. 2024: 11944-11953.

**Questions:**

See weaknesses.

---

> ### Author Response · Authors · 2025-11-25
> **Response to Reviewer fPn9**
>
> # **Response to Reviewer fPn9**
>
> We sincerely thank the reviewer for the insightful comments and suggestions. Below we address each concern in detail.
>
> ---
>
> ## **Weakness 1**
> **Question:**
> *“Please include results for larger initial clean ratios ($ \eta $), e.g., 0.8.”*
>
> **Response:**
> We appreciate this suggestion. In the revised manuscript, we have added additional experiments for $\eta = 0.7$ and $\eta = 0.8$ in **Appendix 10**. The new results continue to confirm our original conclusion: the accuracy benefit contributed by the $1 - \eta$ portion of perturbed data is equivalent to using no more than an additional 20% of clean data. PIL still effectively hinder the model from learning meaningful representations.
>
>
> ---
>
> ## **Weakness 2**
> **Question:**
> *“It would be helpful to include comparisons with more recent work such as CUDA (CVPR’23) and UGE (CVPR’24).”*
>
> **Response:**
> We thank the reviewer for pointing out these recent methods. We have compared our method with CUDA and UGE.
>
>
> ### **Comparison with CUDA (CVPR 2023)**
>
> CUDA is open-sourced, and we use the official implementation. As noted by the original paper, CUDA’s perturbations are generated with large magnitude, often introducing visible distortions.
>
> Because our study focuses on **imperceptible** perturbations (standard $L_\infty \le 8/255$), we additionally introduce **CUDA***—a constrained variant where the perturbation is clamped to an $L_\infty$ budget of 8/255. This enables a fair comparison under commonly adopted perturbation norms.
>
> An excerpt of the results is shown below (full results appear in **Section 4.2.1** and **Appendix 8**):
>
> | Method         | None  | Basic | Rotation | Persp | GrayScale | ChShuf | Cutout | CutMix | MixUp |
> | -------------- | ----- | ----- | -------- | ----- | --------- | ------ | ------ | ------ | ----- |
> | Clean      | 83.93 | 91.45 | 92.05    | 93.38 | 88.84     | 91.99  | 92.71  | 93.49  | 93.87 |
> | CUDA       | **10.35** | 24.74 | 29.53    | 30.70 | 23.74     | 25.72  | 21.62  | 23.95  | 23.54 |
> | CUDA*      | 65.47 | 87.45 | 89.49    | 88.54 | 80.86     | 80.82  | 85.80  | 81.34  | 89.05 |
> | PIL (ours) | 14.70 | **12.87** | **18.15**    | **19.30** | **17.01**     | **10.88**  | **14.62**  | **10.79**  | **11.05** |
>
> CUDA is highly effective when allowed to use large, visible perturbations. However, when restricted to imperceptible perturbation levels (CUDA*), its effectiveness decreases sharply and becomes much less stable across different augmentation settings.
>
> Even when compared directly to the original CUDA, PIL still outperforms it under all augmentation settings except *None*.
>
>
> ### **Comparison with UGE (CVPR 2024)**
>
> UGE's source code is **not publicly released**, so we carefully reimplemented it by following all available details in the original paper. Despite these efforts, our reproduced results do not match the performance reported in it original publication.
>
> | Method         | None  | Basic | Rotation | Persp | GrayScale | ChShuf | Cutout | CutMix | MixUp |
> | -------------- | ----- | ----- | -------- | ----- | --------- | ------ | ------ | ------ | ----- |
> | Clean      | 83.93 | 91.45 | 92.05    | 93.38 | 88.84     | 91.99  | 92.71  | 93.49  | 93.87 |
> | UGE | 83.38 | 91.16 | 90.99    | 91.78 | 88.26     | 90.68  | 92.50  | 92.38  | 91.30 |
>
> UGE achieves only minor accuracy reduction across all augmentation settings in our reimplementation.
>
> To ensure correctness, we contacted the original authors and requested access to their generated unlearnable datasets, but **have not received a response as of now**.
>
> Because of this reproducibility gap, we report UGE in the rebuttal but do not include it in the main paper for now.
>
>
> ### **Runtime Comparison**
>
> For completeness, we also benchmark generation time on CIFAR-10:
>
> | Method         | Time (s) |
> | -------------- | -------- |
> | CUDA       | 9.11     |
> | UGE       | 5043.25  |
> | PIL (ours) | 40.53    |
>
> CUDA, being surrogate-free, can be faster as it avoids the overhead of surrogate models, but its original perturbations are large and performance drops when the perturbation budget is constrained. In contrast, PIL achieves better performance than CUDA in almost all augmentation settings, with only a modest increase in runtime.

---

### Official Review · Reviewer_zTfn · 2025-11-02

**Soundness:** 4
**Presentation:** 3
**Contribution:** 4
**Rating:** 6
**Confidence:** 2

**Summary:**

This paper introduces Perturbation-Induced Linearization (PIL), a method for generating unlearnable examples, which means data samples intentionally perturbed to prevent unauthorized model training. Unlike prior approaches such as REM or TAP that require complex and computationally expensive deep proxy models, PIL generates effective perturbations using only a linear classifier, dramatically improving efficiency.

**Strengths:**

1.The paper provides a comprehensive theoretical analysis of why unlearnable examples work.
2.PIL’s use of a linear classifier as the generator is useful and computationally lightweight.
3.The method can be applied to a broad range of models, which shows the strong generalization.
4.Both experiments and theoretical analysis are comprehensive and clear.

**Weaknesses:**

1.The author's claim that PIL perturbations are ‘imperceptible’ to humans is not quantified by human evaluation results or statistical measurement.
2.The authors empirically show that gradients from unlearnable samples are nearly orthogonal to those from clean samples, implying training interference. However, this is discussed qualitatively. A more rigorous analysis could include “quantitative measures such as cosine similarity distributions between gradient vectors per class.”

**Questions:**

1.Could authors provide a more rigorous analysis about the quantitative analysis between unlearnable samples and clean samples?
2.Have the authors considered including the kernelized linearization to capture the richer structure compared with only a single-layer linear classifier?
3.Could authors measure Jacobian singular value spectra or Hessian eigenvalue distributions before and after PIL perturbations?

---

> ### Author Response · Authors · 2025-11-25
> **Response to Reviewer zTfn**
>
> # **Response to Reviewer zTfn**
>
> We sincerely thank the reviewer for the insightful comments and suggestions. Below we address each concern in detail.
>
> ---
>
> ## **Weakness 1**
>
> **Question:**
> *“The author's claim that PIL perturbations are ‘imperceptible’ is not quantified by human evaluation results or statistical measurement.”*
>
> **Response:**
> We appreciate the reviewer’s suggestion. In the revised manuscript, we have added a discussion of the perturbation magnitude and its perceptual impact.
>
> 1. **Perturbation Norm Discussion (added in Section 4):**
>    We adopt an **8-pixel perturbation budget**, a widely used standard in adversarial robustness research, where perturbations of $\le 8/255$ are generally considered *imperceptible to human observers*. We now explicitly discuss this point and provide appropriate citations in the revised paper.
>
> 2. **Perceptual Quality Metrics (added in Appendix 13):**
>    To further support the imperceptibility claim, we compute **PSNR** and **SSIM** between the clean dataset and the PIL-perturbed dataset. We provide the PIL results here for quick reference:
>
>    | Dataset Comparison | PSNR     | SSIM     |
>    | ------------------ | -------- | -------- |
>    | Clean vs. PIL      | **30.2** | **0.95** |
>
>    For additional context, we compare with commonly used JPEG compression levels:
>
>    | JPEG Quality | PSNR | SSIM |
>    | ------------ | ---- | ---- |
>    | 90           | 33.8 | 0.97 |
>    | 80           | 31.3 | 0.95 |
>    | 70           | 29.9 | 0.93 |
>
>    These results indicate that PIL-perturbed images exhibit perceptual quality closely aligned with that of **JPEG 80**, a compression level at which visual differences from the original image are typically subtle and often difficult for human observers to detect.
>
> ---
>
> ## **Weakness 2 & Question 1**
>
> **Question:**
> *“A more rigorous analysis could include “quantitative measures such as cosine similarity distributions between gradient vectors per class. Could authors provide a more rigorous analysis about the quantitative analysis between unlearnable samples and clean samples?”*
>
>
> **Response:**
> We agree and have added a new quantitative gradient similarity analysis in the **appendix 4.2**.
>
> Specifically, we compute **intra-class gradient cosine similarity distributions** for models trained on clean and PIL-perturbed datasets.
> Our findings show that:
>
> > **PIL samples consistently induce higher intra-class gradient similarity**, indicating *collapsed and less informative gradient directions*.
>
> This supports our hypothesis that PIL interferes with the gradient flow during training and provides a rigorous quantitative complement to the qualitative discussion.
>
> Detailed numerical results and plots are now included in the appendix of the revised paper.
>
> ---
>
> ## **Question 2**
> **Question:**
> *“Have the authors considered including the kernelized linearization to capture the richer structure compared with only a single-layer linear classifier?”*
>
> **Response:**
> Yes, we have explored richer structures beyond a single-layer linear classifier. We tested two extensions:
>
> 1. **Introducing a hidden layer with nonlinear activations**
> 2. **Using an RBF kernel**
>
> The classification accuracy (on clean CIFAR10 test set) is summarized below:
>
> | Base Model                 | Accuracy (%) |
> | -------------------------- | ------------ |
> | Linear (used in our paper) | **12.87**    |
> | 2-layer MLP                | **20.84**    |
> | RBF Kernel Model           | **16.31**    |
>
> Both directions increase the model’s capacity and allow it to capture more complex features.
> However, **this is contrary to the design goal of PIL**, which aims to create *simple shortcuts* that prevent the model from learning meaningful features in unauthorized training.
>
> As shown above, more nonlinear models actually **reduce the protective effect of PIL**, confirming our design intuition and justifying our focus on linear models.
>
> ---
>
> ## **Question 3**
>
> **Question:**
> *“Could the authors measure Jacobian singular value spectra or Hessian eigenvalue distributions before and after PIL?”*
>
> **Response:**
> We thank the reviewer for the excellent suggestion. We have added new analysis of the **Jacobian singular value spectra**.
>
>
> We compute the Jacobian singular value spectra on the clean CIFAR-10 test set for two ResNet-18 models:
>
> 1. Trained on clean training data
> 2. Trained on PIL-perturbed training data
>
> For each of the 10,000 test images, we compute the singular values (sorted descending) and average them across the dataset.
>
> **Key Findings:**
>
> * Singular values **increase across all modes** after applying PIL.
> * Largest singular value: **69.03 → 72.76**
> * Smallest singular value: **0.334 → 0.799**
>
> This suggests:
>
> > PIL increases the model’s output sensitivity along many input directions, distributing sensitivity more evenly and increasing the uncertainty in class selection.
>
> A detailed figure and discussion have been included in the revised **Appendix 11**.

---

### Author Response · Authors · 2025-12-02
**Rebuttal Overview**

Dear ICLR PC/SAC/AC/Reviewers,

Thank you very much for your tremendous efforts during this challenging and unexpected situation. We are especially grateful to the Area Chairs for taking on additional responsibilities to ensure fairness. Your work has been invaluable, and we sincerely appreciate it.

To assist your evaluation, we provide a brief summary of the reviewers' comments and our responses.

We are grateful that the reviewers recognized the efficiency and performance of our proposed PIL method, the novelty of our findings, and the clarity of our writing and presentation.

All reviewers provided constructive and helpful suggestions that have helped make our work clearer and easier to understand. Many concerns raised by reviewers can be addressed through additional experiments, which we have conducted. These experiments fall into several categories:
1. quantitative metrics for visual discrepancy,
2. evaluation of whether to adopt more complex surrogate models,
3. more detailed gradient analysis (such as intra-class cosine similarity and Jacobian singular value spectra),
4. new results for larger initial clean ratios to validate how perturbed samples contribute to accuracy improvements,
5. broader comparisons with additional baselines, and
6. more experiments on countermeasures (such as JPEG compression).

For **analytical experiments**, the new results continue to **support our original conclusions and statements**. For **comparative experiments**, the new results further **demonstrate PIL's effectiveness**, such as its robustness to JPEG compression.

The concerns raised by Reviewers zTfn and fPn9 can be addressed through these additional experiments. Reviewers VbQk and QLdv both pointed out issues regarding Section 6, which we believe stemmed from the original space limitations. We have now expanded this section with a more detailed discussion, which we believe addresses these concerns.

We also engaged in discussions with Reviewer VbQk regarding whether linearly separable datasets lead to linear deep models and how to measure the linearity of deep models. These are interesting questions that we have carefully considered, and we have provided detailed responses exploring additional linearity proxies and addressing these questions.

We have also shared with Reviewer QLdv our design considerations for the PIL algorithm and related comparative experiments, which we hope address concerns about our method's motivation and design details.

Finally, we would like to emphasize a key point regarding the contribution of our work: our work not only delivers **comparable or better performance** with **orders-of-magnitude lower computational cost**, but also **reveals a key mechanism underlying unlearnable examples**: inducing deep models to behave more like linear models. We believe this mechanism-level perspective will offer useful insights for future research on unlearnable examples, and hope this addresses Reviewer QLdv's concern about the contribution of our work.

For clarity, we have created **a summary table (see next comment)** that outlines each reviewer's concerns and our responses.

We summarize the timeline of our rebuttal process for clarity:
- On **25 Nov 2025**, we completed all additional experiments requested by the reviewers, updated the manuscript, and submitted detailed comments addressing each question.
- On **30 Nov 2025** and **1 Dec 2025**, we submitted further refinements to both the manuscript and our responses, aiming to make the work clearer and more accessible given the unexpected challenges affecting the review process.

Thank you again for your time, understanding, and dedication. We deeply appreciate the extraordinary effort from the organizing committee, and we remain grateful for the opportunity to have our work assessed during such unforeseen and difficult conditions.

Warm regards,

Authors of submission7899

---

> ### Author Response · Authors · 2025-12-02
> **Summary of Reviewer Concerns and Our Responses**
>
> | Reviewer | Concern | Our Solution |
> |----------|---------|--------------|
> | zTfn | PIL perturbations' imperceptibility lacks quantitative metrics | Added PSNR and SSIM metrics (Appendix 13), added perturbation norm discussion (Section 4) |
> | zTfn | Need more rigorous gradient analysis (gradient similarity distributions) | Added intra-class gradient cosine similarity distributions analysis (Appendix 4.2) |
> | zTfn | Whether to consider kernelized linearization to capture the richer structure | Tested 2-layer MLP and RBF kernel models, verified that linear models better serve PIL's design goal |
> | zTfn | Whether to measure Jacobian singular value spectra | Added Jacobian singular value spectra analysis (Appendix 11) |
> | fPn9 | Need results for larger initial clean ratios ($\eta=0.8$) | Added results for $\eta=0.7$ and $\eta=0.8$ (Appendix 10) |
> | fPn9 | Need comparisons with CUDA [1] and UGE [2] | Added comparisons with CUDA and UGE, including performance and runtime comparison (Section 4.2.1, Section 4.3, Appendix 8)|
> | VbQk | JPEG compression evaluation (different quality levels) | Added JPEG compression experiments, systematically evaluating different quality factors (Table 3, Section 4.2.1) |
> | VbQk | Whether orthogonal projection would work against PIL | Conducted orthogonal projection experiments and verified that PIL remains effective under orthogonal projection |
> | VbQk | Claim about partial dataset perturbation already shown in Huang et al. [3] | Explicitly acknowledged Huang et al.'s finding, emphasized our original contribution of gradient orthogonality phenomenon (Appendix 4.1, Table 10), and quantitative experiments (Appendix 10.2, Fig. 4). Rewrote and expanded Section 6 to improve clarity |
> | VbQk | Other linearity proxies, and whether linearly separable data leads to linear models | Explored additional linearity proxies ($R^2_\mathrm{global}$ and $R^2_\mathrm{dir}$), discussed relationship between linearly separable data and linear models |
> | VbQk | Need ablation study | Added ablation experiments focusing on the cases $\lambda = 0$ and $\lambda = 1$ (Appendix 12), highlighting the effects of each component. |
> | VbQk | SEP's smaller FGSM gap | Softened wording, guided readers to focus on the key finding that all entries are positive (Section 5)|
> | QLdv | Motivation and mechanism unclear, confusion about $\delta$ and $\delta_1+\delta_2$ combination | Clarified motivation and mechanism, explained design of two components, conducted PIL$^+$ comparison experiments to validate sign choice |
> | QLdv | Method not consistently best, limited contribution | Emphasized two key contributions: efficiency advantage (orders-of-magnitude lower cost) and mechanism discovery (inducing linearity) |
> | QLdv | Theorem 1 self-contradictory (large $\alpha$ leads to accuracy drop) | Rewrote and expanded Section 6, clarified figure labeling, explained that accuracy drop is due to data scarcity rather than failure of Theorem 1 |
>
> ---
>
> [1] Vinu Sankar Sadasivan, Mahdi Soltanolkotabi, and Soheil Feizi. Cuda: Convolution-based unlearnable datasets. In Proceedings of the IEEE/CVF Conference on Computer Vision and Pattern Recognition, pages 3862–3871, 2023.
>
> [2] Ye J, Wang X. Ungeneralizable examples[C]//Proceedings of the IEEE/CVF Conference on Computer Vision and Pattern Recognition. 2024: 11944-11953.
>
> [3] Unlearnable Examples: Making Personal Data Unexploitable. Huang et al., 2021

---

### Meta-Review · Area_Chair_qxyg · 2026-01-02

**Summary:**

The paper proposes a method to generate unlearnable examples using a linear surrogate model, which achieves comparable or better performance than existing surrogate-based methods while reducing computational time significantly. Although the experiments are not very thorough, this is a value addition to the literature of AI security.

**Reviewer Concerns:**

1. The proposed method is not consistently best and shows incremental improvements.
2. The paper primarily focuses on methods developed up to 2022.

**Reviewer Scores:**

zTfn would keep the score.
fPn9 would keep the score.
VbQk would keep the score.
QLdv would keep the score.

---

### Decision · Program_Chairs · 2026-01-26

Accept (Poster)